# SPARTUN3D: SITUATED SPATIAL UNDERSTANDING OF 3D WORLD IN LARGE LANGUAGE MODELS

**Yue Zhang**[1]  **Zhiyang Xu**[2]  **Ying Shen**[3]  **Parisa Kordjamshidi**[1*]  **Lifu Huang**[4*]
[1]Michigan State University  [2]Virginia Tech
[3] University of Illinois at Urbana-Champaign  [4]UC Davis
zhan1624@msu.edu, zhiyangx@vt.edu, ying22@illinois.edu

## ABSTRACT

Integrating the 3D world into large language models (3D-based LLMs) has been a promising research direction for 3D scene understanding. However, current 3D-based LLMs fall short in situated understanding due to two key limitations: 1) existing 3D datasets are constructed from a global perspective of the 3D scenes and lack situated context. 2) the architectures of existing 3D-based LLMs lack explicit alignment between the spatial representations of 3D scenes and natural language, limiting their performance in tasks requiring precise spatial reasoning. We address these issues by introducing a scalable situated 3D dataset, named *Spartun3D*, that incorporates various situated spatial reasoning tasks. Furthermore, we propose *Spartun3D-LLM*, built on an existing 3D-based LLM but integrated with a novel situated spatial alignment module, aiming to enhance the alignment between 3D visual representations and their corresponding textual descriptions. Experimental results demonstrate that both our proposed dataset and alignment module significantly enhance the situated spatial understanding of 3D-based LLMs.

## 1 INTRODUCTION

Recent advances in large language models (LLMs) have demonstrated their remarkable reasoning and communication capabilities across various tasks and modalities (Achiam et al., 2023; Alayrac et al., 2022; Zhang et al., 2023a; Rubenstein et al., 2023). Building on these breakthroughs, there has been a growing interest in extending LLMs to the 3D world (3D-based LLMs) (Hong et al., 2023; Huang et al., 2023; Chen et al., 2024; Zhen et al., 2024; Wang et al., 2023). Existing studies mainly focus on integrating various 3D scene representations into LLMs, enabling the models to perform 3D grounding and spatial reasoning through natural language. For example, 3D-LLM (Hong et al., 2023) utilizes multi-view images to represent 3D scenes, pioneering a new direction in this field, while LEO (Huang et al., 2023) further pushes the boundary by directly injecting 3D point clouds into LLMs, aiming to develop a generalist embodied agent capable of 3D grounding, embodied reasoning, and action planning.

**Question**: what should you do to wash hands?

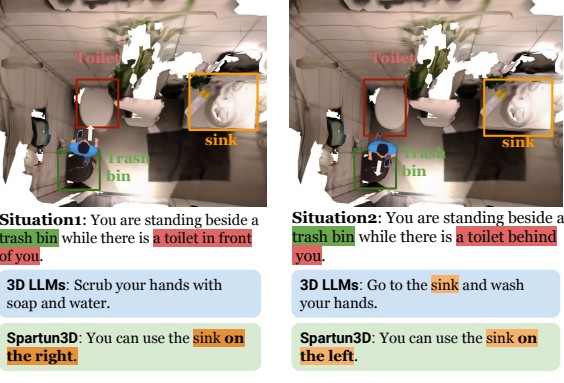

Figure 1: Illustration of situated scene understanding of Spartun3D-LLM compared to other 3D-based LLMs.

Despite the promising progress, current 3D-based LLMs still fall short in ***situated understanding***, a fundamental capability for completing embodied tasks, such as Embodied Question Answering (Das et al., 2018), Vision and Language Navigation (Anderson et al., 2018; Zhang et al., 2024c), robotic manipulation (Shridhar et al., 2022), and many others. Situated understanding refers to the ability to

---

*Co-supervision.

**Situated Captioning**

**Situation:** *Standing beside **blue octagon wide bed** that is messy while there is a **window on the left**.*

**Situated QA**

**Q**: Describe the scene from your current position.
**A**: **In front**, there's a rectangular box and a big picture that can be hung or moved. Nearby, there's a tall nightstand beside the bed. **To the right**, there's a tall pillow close to another pillow and a blanket. **Behind**, a polished mirror and a flexible bag are near a closed door with hanging clothes. **On the left**, there's an artificial lamp and a desk close to a sofa chair and trash bin, a curtain, another lamp, and a tall wardrobe near a cabinet.

Other Views

*Object Attribute and Relation*

**Q**: Where is the desk located?
**A**: **Behind you**

*Object Affordance*

**Q**: I need to check my appearance. Where should I go?
**A**: You should go to the **mirror behind you**.

*Situated Planning*

**Q**: I want to make the room brighter and then sit down to relax. What should I do?
**A**: **Turn to your left** and head towards the window. Open the curtains to let in more natural light. After adjusting the curtains, head towards the **sofa chair on your left side** to sit and relax. You may need to **pass other chairs and a desk** in your path.

Figure 2: Examples of Spartun3D. Green box and arrow show the standing point and orientation.

interpret and reason about a 3D scene from a ***dynamic egocentric perspective***, where the agent must continuously adjust understanding based on its changing position and evolving environment around it. This capability is crucial because an agent's reasoning and response to the same question can vary depending on its current situation. For example, as shown in Fig 1, given the same question *"What should you do to wash hands?"*, the agent might need to answer *"use the sink on the left/right"* based on the agent's current perspective and location relative to the *"sink"*.

However, achieving such situated understanding remains challenging for current 3D-based LLMs, and we identify two primary reasons. **First**, most existing 3D datasets (Chen et al., 2021; Azuma et al., 2022; Zhu et al., 2023; Huang et al., 2023) are constructed from a global perspective of 3D scenes, lacking the situated information necessary for training models to reason from an agent's perspective. As a result, models fine-tuned on these datasets struggle to develop situated reasoning ability. While the introduction of SQA3D (Ma et al., 2022) has made progress by providing a situated 3D dataset, the dataset is mainly human-annotated, making it expensive and difficult to scale for the large-scale training needed by 3D-based LLMs. **Second**, existing 3D-based LLMs inject 3D representations into LLMs by simply concatenating tokens from different modalities (*e.g.*, text, images, 3D point clouds). While this allows for basic cross-modal interaction, it lacks an explicit mechanism to align the situated spatial information from the 3D scene with natural language. Therefore, the models struggle to capture the critical spatial relationships for situated understanding.

To address the aforementioned issues, we propose two key innovations: we first introduce a scalable, LLM-generated dataset named ***Spartun3D***, consisting of approximately 133k examples. Different from datasets used by previous 3D-based LLMs (Zhu et al., 2023; Huang et al., 2023; Hong et al., 2023), Spartun3D incorporates various situated spatial information conditioned on the agent's standing point and orientation within the environment, consisting of two situated tasks: situated captioning and situated QA. Situated captioning is our newly proposed task that requires generating descriptions of the surrounding objects and their spatial direction based on the agent's situation. Situated QA is designed with different types of questions targeting various levels of spatial reasoning ability for embodied agents. Furthermore, based on Spartun3D, we propose a new 3D-based LLM, ***Spartun3D-LLM***, which is built on the most recent state-of-the-art 3D-based LLM, LEO (Huang et al., 2023), but integrated with a novel *situated spatial alignment* module that explicitly aligns 3D visual objects, their attributes and spatial relationship to surrounding objects with corresponding textual descriptions, with the goal of better bridging the gap between the 3D and text spaces.

We conduct extensive experiments across a variety of tasks, including Spartun3D, SQA3D, and MP3D Nav (Savva et al., 2019). Our results demonstrate that our model, trained on Spartun3D, exhibits strong generalization to other tasks, specifically in zero-shot settings, highlighting the effectiveness of our proposed dataset. Additionally, Spartun3D-LLM outperforms the baseline on nearly all tasks, indicating that incorporating direct text supervision improves the model's spatial understanding ability. This is further supported by our observation that the explicit alignment module improves the generation of more fine-grained, context-aware spatial information.

## 2   RELATED WORK

**Situated Scene Understanding** is essential for various embodied tasks, including embodied QA (Das et al., 2018; Wijmans et al., 2019), vision-and-language navigation (Anderson et al., 2018;

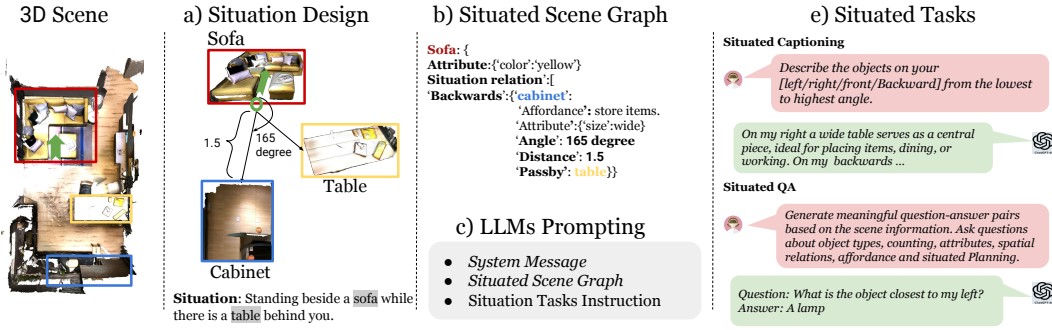

Figure 3: Illustration of Spartun3D Dataset Construction Process. Given a 3D scene: a) we first select a pivot object (*e.g.,* sofa) and a referent object (*e.g.,* table) to define the situation; b) we create situated scene graph based on situation, incorporating various spatial relationships (See Fig. 5); c) we use the scene graph to prompt GPT-4o (shown in gray box) to generate data; e) we utilize different prompting strategies for generating situated captions and QA pairs.

Zhang et al., 2021; Zhang & Kordjamshidi, 2022a;b; 2024) and robotic manipulation (Shridhar et al., 2022; Jiang et al., 2022; Driess et al., 2023). Recently, SQA3D (Ma et al., 2022) introduces a human-annotated dataset where the model generates answers based on questions and given situations. SIG3D (Man et al., 2024) highlights the situated awareness via a situation-grounded 3D VL reasoning architecture. While SQA3D is crucial in 3D vision-language learning (Zhu et al., 2023; Huang et al., 2023), its reliance on human annotations makes it costly and difficult to scale for the large-scale training required by 3D-based LLMs. In contrast, we use the LLM to design an automated pipeline to generate situated questions and answers, enhancing both the scalability and diversity in data collection procedure.

**Grounded 3D Scene Understanding.** Compared to 2D vision-language tasks (Song et al., 2024; Antol et al., 2015; Zhang et al., 2024a; Guo et al., 2024a;b; Xu et al., 2024; Wang et al., 2024; Qi et al., 2024; Guo et al., 2025), 3D scenes introduce additional dimensions of knowledge that is more challenging to align with text modalities. Early studies in this area primarily focused on grounding language to individual objects within 3D environments (Achlioptas et al., 2019; 2020; Chen et al., 2020; 2019; 2021). Recently, 3D Vista (Zhu et al., 2023) proposes a pre-trained VL Transformer for 3D vision and text alignment, and a few works utilize LLMs in understanding 3D scenes (Hong et al., 2023; Huang et al., 2023; Chen et al., 2024; Zhen et al., 2024; Yang et al., 2024). However, these works are less effective in situated understanding tasks that require generating answers from the agent's dynamic perspectives. MSQA (Linghu et al., 2025) is a concurrent work that also explore situated understanding of 3D-based LLMs. However, our approach differs in both dataset construction and model design. While MSQA focuses on leveraging images to assist in describing a situation, our method emphasizes directly understanding the situation from textual descriptions, leading to distinct contributions in the field.

## 3 SPARTUN3D DATASET CONSTRUCTION

To better equip 3D-based LLMs with the capability of understanding situated 3D scenes, we introduce Spartun3D, a diverse and scalable situated 3D dataset. To ensure the scalability of Spartun3D, we carefully design an automatic pipeline that leverages the strong capabilities of GPT-4o (OpenAI, 2024), with three key stages as shown in Fig. 3: (1) Designing diverse situations that specify the agent's standing point and orientation given a 3D scene as input (Sec. 3.1); (2) Constructing situated scene graphs to describe the spatial relationships between the agent and objects in the environment conditioned on the agent's situations (Sec. 3.2) ; and (3) Prompting LLMs to generate dataset based on situated scene graphs (Sec. 3.3).

### 3.1 SITUATION DESIGN

The 3D scenes in Spartun3D are taken from 3RScan (Wu et al., 2021), which provides a diverse set of realistic 3D environments. Given a particular 3D scene with all the objects labeled by humans from 3RScan, such as the example shown in Fig. 3, our first step is to generate diverse situations for the agent. To construct the situation, we begin by identifying the standing point and orientation and

then complete a situation description accordingly using the following template: "*You are standing beside {pivot object name}, and there is {referent object name} on the {left/right/front/backward}.*" The elements within {} specify the key components that together define the situation. Below, we define the agent's standing point and orientation and explain how these elements are obtained to construct diverse and reliable situations.

**Standing Point and Orientation.** We begin with determining the agent's standing point and orientation within the 3D scene. Our approach is to place the agent beside an object, ensuring a clear reference for orientation when interacting with the environment. Specifically, we project all objects from 3D space onto a 2D plane, focusing only on the x and y coordinates to construct a bird-eye-view of the scene. From this projected 2D scene, we randomly select an object from the set of segmented objects within the 3D scene. To ensure the constructed situation remains realistic, we exclude objects that are positioned too high to avoid unnatural situations like *"standing beside the lamp on the ceiling"*. As a result,

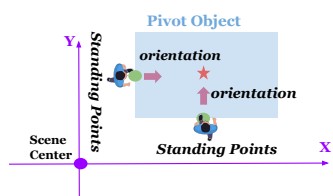

Figure 4: Standing Point and Orientation Selection.

we limit the selection to objects whose z-axis is below the average height of all objects in the scene. Then, we choose a midpoint from two sides of the selected object's bounding box that are closest to the center of the scene, as shown in Fig. 4. By prioritizing the side closest to the center, we minimize this risk and keep the agent within the scene's boundaries. Finally, the selected midpoint will be used as the agent's standing point. In addition, we need to determine the agent's orientation. We assume the agent's orientation is always facing forward to the center of the selected object. This guarantees that the selected object remains within the agent's field of view.

**Pivot and Referent Object.** Once the agent's standing point and orientation are determined, we refer to the object that the agent stands beside as the *"pivot object"*, and other objects surrounding the pivot object are potential referent objects. A referent object is then randomly selected, and its relative position (left/right/front/backward), with respect to the agent's standing point and orientation, is used to generate the description of the situation.

## 3.2 SITUATED SCENE GRAPH CONSTRUCTION

Building on the agent's situation, we further construct a situated scene graph that captures the comprehensive spatial relationships between the agent and its surrounding objects. Existing 3D-based LLMs (Zhu et al., 2023; Huang et al., 2023) represent scenes in a structured manner using JSON-formatted scene graphs, including detailed scene context of object attributes and relative spatial relationships between objects. However, their spatial relations are based on a global view, such as a bird-view-eye perspective (as shown in Fig. 5). To enable situated understanding, we introduce a *situated-scene-graph* adapted from the original global scene graph to capture all relative spatial relationships between the agent's standing point and surrounding objects as follows:

- *Rotation Angles.* We calculate rotation angles that reorient the agent from its orientation to the surrounding objects. Specifically, we first calculate the horizontal angle between the standing point and the center of the pivot object. Next, we calculate the horizontal angle between the standing point and a surrounding object. The rotation angle is determined by the difference between these two angles. We further normalize the rotation angles such that larger values correspond to a greater degree of rightward rotation.

- *Direction.* We classify the object's rotation angles to the agent into four directional categories according to a predefined standard: [*front, right, backward, left*] (see Fig. 5 (b)). For instance, an object is categorized as "*right*" if the turn angle falls within the range of [45-135] degrees relative to the agent's forward-facing orientation.

- *Distance.* We compute the Euclidean distance between the agent's standing point and the center of the bounding boxes of surrounding objects.

- *Passby Objects.* We assess whether the agent can move freely from its standing point to other objects. We draw a straight line from the agent's standing point to the center of the referenced object. If this line intersects any other objects in the scene, those objects are considered "passby objects". For example, as illustrated in Fig 5 (d), the "*table*" is a passby object between the agent and the "*kitchen cabinet*". We explictly include the information of passby objects to help the agent build awareness of objects that might influence its path while navigating.

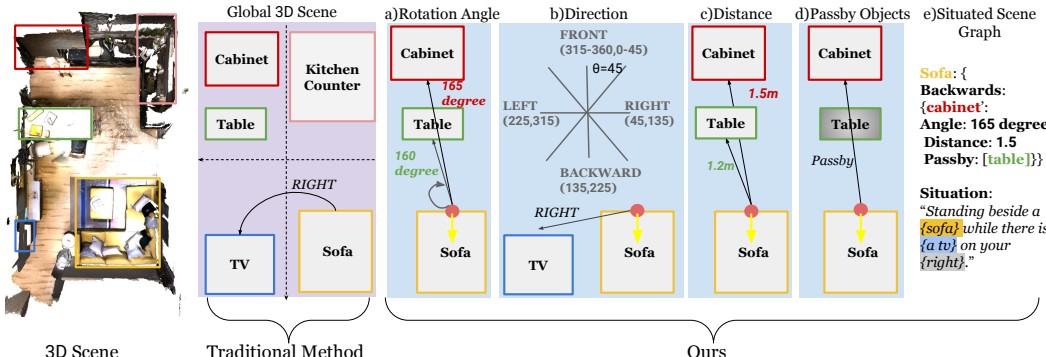

Figure 5: Spatial information in Situated Scene Graph. The red dot and green arrow show the standing point and orientation, respectively. In this example, the pivot object is the "sofa", the referent object is the "TV", and the surrounding objects include the "table" and "cabinet".

After gathering the spatial information described above, we organize it into a JSON format (shown in Fig. 5 (e)), which is then used as input to prompt LLMs to generate our datasets.

## 3.3 LLMS PROMPTING

We design specific instructions to prompt **GPT-4o** (OpenAI, 2024) for two situated tasks: **Situated Captioning** and **Situated QA**. For both tasks, we ask GPT-4o to provide responses considering situated spatial information. Detailed prompts for each task are provided in Tab. 10 in the Appendix, and examples of the generated dataset are shown in Fig. 2.

**Situated Captioning** is our newly introduced task, aiming to generate brief situated descriptions of the surrounding objects as the agent performs a 360° clockwise rotation starting from its standing point and orientation. The motivation for introducing this task stems from its crucial role in embodied tasks, such as navigation, where the agent must interpret and reason about its environment from 360° panoramic views to make decisions about movement and interaction (Zhou et al., 2024; Zhang & Kordjamshidi, 2023; Zhang et al., 2024b). Therefore, we guide GPT-4o to generate descriptions progressively, starting from lower rotation angles and moving toward higher angles in each direction.

**Situated QA.** We design three types of questions for the Situated QA task, each targeting a different aspect of spatial reasoning for embodied agents. Unlike previous works that rely on a single generic prompt for all question types, we develop tailored prompting strategies for each question type, encouraging LLM to generate QA pairs focusing on different levels of reasoning.

- *Object Attribute and Relations* include questions about objects attributes, such as *color*, *shape*, and *size*, while also incorporating situated spatial information. For instance, the questions to identify *"the color of the table positioned to the left"*, and determine *"how many pictures are hanging on the wall to the right"*.

- *Object Affordance* focuses on the function utility of the objects, often based on common sense knowledge about how objects are used. Similarly, we require situated spatial information to be part of the answer. For example, when asked *"Where can you check your appearance?"*, the correct answer should be *"mirror on your left"*, specifying both the object name (*mirror*) and its spatial location from the agent.

- *Situated Planning* is the most challenging task, as it requires the agent to perform multi-hop situated spatial reasoning. The agent must not only recognize its surroundings but also plan and execute a series of actions across multiple steps, where each subsequent action depends on the outcome of the previous one. In our dataset, we implement 2-hop reasoning, which requires the agent to perform a sequence of two continuous actions. For example, given the example in Fig. 2, *"make the room brighter and then sit down to relax."*, the agent needs to first turn left from its orientation to face and move toward the window, open it to brighten the room, then based on its new position, the agent continues turning left toward the sofa chairs and sits down.

Figure 6: Human Evaluation of Spartun3D.

## 3.4 DATASET STATISTICS AND QUALITY EVALUATION

In total, we collect approximately **10k** situated captions and **123k** QA pairs. For the tasks of object attribute and relation and the tasks of affordance, we sampled around 10 situations per scene. For captioning and planning tasks, we sample around 5 situations per scene due to the increasing cost of longer token sequences required for these tasks. For each task, we split the data instances into a training and test set. Table 1 shows the statistics of our dataset.

We conduct human evaluation to assess the quality of Spartun3D, introducing scores based on two key criteria: *language naturalness*, which evaluates whether the text reads as if it were naturally written by a human, and *spatial fidelity*, which ensures that the data accurately reflects the 3D scene with correct spatial relationships. Detailed explanations of the criteria are in Sec. A.1. Each criterion is rated on a scale from 1 to 5, and the average of these two

Table 1: Dataset statistics of Spartun3D and human validation results.

| Tasks | # of Examples | Train/Test |
|---|---|---|
| Captioning | $\sim 10K$ | $8,367/1,350$ |
| Attr. & Rel. | $\sim 62K$ | $61,254/8,168$ |
| Affordance | $\sim 40K$ | $35,070/5,017$ |
| Planning | $\sim 21K$ | $19,434/2,819$ |

scores is the overall human score. We randomly select 50 examples from each task and compute human scores of situation, question, and answer, respectively. As shown in Fig. 6 (a), the average scores align with the complexity of each task, with relatively lower scores for captioning and planning tasks. To evaluate how our generated data compares to human-annotated data, we sampled 50 examples from SQA3D and combined them with our dataset. Our data shows a similar trend in human evaluation results across different question types as observed in SQA3D (shown in Fig. 6 (b)). We finally evaluate how different prompting strategies influence the quality of the data. We experiment with two types of prompts for representing spatial information to prompt GPT-4o: **Cord-prompt**, which consists of object center coordinates, standing point, orientation, and instructions for calculating distances and rotation angles, and **Spa-prompt**, consisting of the calculated angles and distance based on the approaches we described in Sec. 3.3. An example of each type of prompt can be found in Tab. 11. Fig.6 (c) shows the percentage of examples with high human scores ($\geq 4$) for each prompt across tasks. The results indicate that Cord-prompt yields unsatisfactory results, revealing that LLMs lack strong 3D spatial reasoning when interpreting raw spatial coordinates, which is consistent with their struggles in spatial reasoning across text and 2D images (Zhang et al., 2024d; Premsri & Kordjamshidi, 2024; 2025). Our Spa-prompt significantly improves the quality of the generated dataset by providing qualitative spatial relations (*e.g.* distance, direction).

## 4 MODEL ARCHITECTURE

In addition to enhancing the situated understanding of 3D-based LLMs with Spartun3D, we also propose a new 3D-based LLM, named **Spartun3D-LLM**, which integrates a novel *Situated Spatial Alignment* module to strengthen the alignment between the situated 3D visual features and their corresponding textual descriptions. Spartun3D-LLM is built upon LEO (Huang et al., 2023), which represents the most recent and state-of-the-art 3D-based LLM, and directly takes 3D point cloud data as input, making it well-suited for spatial reasoning tasks in 3D environments. Fig. 7 illustrates the overview architecture of Spartun3D-LLM.

### 4.1 BACKGROUND

**Problem Formulation.** We formally define the input as a triple $< C, S, Q >$, where $C$ is the 3D scene context, $S$ is the situation, and $Q$ is a question. The situation $S$ can be further denoted as $S = < S^t, S^p, S^r >$, where $S^t$ is a textual situation description, and $S^p$ and $S^r$ are the standing

points and orientation, respectively. Specifically, $S^p$ is a 3D coordinate in the form $<x, y, z>$ and $S^r$ is the quaternion $<qx, qy, qz, w>$, where $<qx, qy, qz>$ is the rotation axis and $w$ is the rotation angle. For simplicity, we define $z = 0$ to calculate the rotation angle on a 2D plane. The task is to generate a textual answer, denoted as $A$, given scene context $C$, situation $S$, and question $Q$. During training, $S^p$ and $S^r$ are provided to the agent to rotate and translate the environment, while during testing, only questions and situations are provided.

**Backbone.** LEO takes text, 2D image (optional), and 3D point clouds as input and formulate comprehensive 3D tasks as autoregressive sequence generation. Specifically, data from different modalities are converted into a sequence of tokens as input to the LLM. The text tokens include system messages (*e.g., "You are an AI visual assistant situated in a 3D scene."*), situations, and questions. These tokens are then embedded into vector representations using an embedding look-up table. For 3D point clouds, LEO first applies seg-

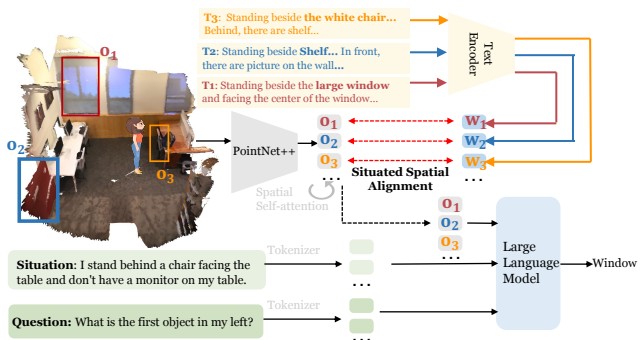

Figure 7: Spartun3D-LLM Model Architecture.

mentation masks to extract the point clouds of individual objects in the 3D scenes. Then, the sampled points of each object are input into a object-centric point cloud encoder, PointNet++ (Qi et al., 2017) pre-trained on ScanNet (Dai et al., 2017), to obtain the object-level representations.

Formally, we denote the representation of input text tokens as $\mathbf{W} = [\mathbf{w}_1, \mathbf{w}_2, ..., \mathbf{w}_M] \in \mathbb{R}^{M \times D}$, where $M$ denotes the number of input tokens, and $D$ represents the dimensionality of each token's embedding. Additionally, the input object visual representations are expressed as $\mathbf{O} = [\mathbf{o}_1, \mathbf{o}_2, ..., \mathbf{o}_K] \in \mathbb{R}^{K \times D}$, where $K$ is the number of extracted objects from the scene. Finally, the output answer are represented as $\mathbf{A} = [\mathbf{a}_1, \mathbf{a}_2, ..., \mathbf{a}_N]$, where $N$ is the number of tokens in the response. The model's objective is to generate the answer given these combined inputs. The loss for generating the $i$-th token of the output answer is formulated as follows:

$$\mathcal{L}_{\text{LM}}(\theta) = \sum_i log p_\theta(\mathbf{a}_i | \mathbf{a}_{i-1}, \mathbf{W_S}, \mathbf{o}). \tag{1}$$

LEO can integrate various LLM backbones, including OPT1.3B (Zhang et al., 2023b) and Vicuna-7B (Chiang et al., 2023). In our experiments, we fine-tune LEO with different LLM backbones on our proposed dataset via LoRA (Hu et al., 2021).

## 4.2 SITUATED SPATIAL ALIGNMENT MODULE

Situated tasks require robust spatial reasoning abilities to comprehend the position, orientation, and spatial relationships of objects within a 3D environment. Existing 3D-based LLMs typically process inputs by concatenating output representations from various modality encoders. While this method facilitates the integration of data across different modalities, it does not inherently ensure that the 3D visual representations encode situated spatial information or effectively align with textual descriptions, which potentially limits the model's ability to perform tasks that require precise spatial understanding. To tackle this challenge, we introduce a novel *Situated Spatial Alignment Module* to improve the alignment between the object-centric 3D visual representations and their situated textual descriptions. The process begins by generating detailed situated textual descriptions for each object. Subsequently, an alignment loss is introduced, which directs the model in effectively learning the 3D visual representations based on these situated textual descriptions.

**Situated Textual Descriptions.** For each object, we construct a comprehensive situated textual description based on a template that captures the object's name, attributes, and spatial relations with nearby objects, as *"Stand besides {object name} and facing the center of the {object name}, in front, there are {a list of nearby objects}; on the right, ...; behind ...; and on the left..."*. The object's attributes are also considered (e.g., *"white chair"*). We consider up to five objects per direction. If no object is present in a specific direction, the description explicitly states this, ensuring to provide complete information about the 3D environment.

Table 2: Experimental Results on Spartun3D Situated QA Tasks. ∗ represents the model initialized with LEO instruction-tuned weights. [Keys: C: CIDER; B-4: BLEU-4; M: METEOR; R: ROUGE; Sim: Sentence Similarity; EM: Exact Match; **Bold: best results**].

| Models | LLMs | Attributes and Relations | | | | | Affordance | | | | | Situated Planning | | | | |
|---|---|---|---|---|---|---|---|---|---|---|---|---|---|---|---|---|
| | | C | B-4 | M | R | EM | C | B-4 | M | R | S | C | B-4 | M | R | S |
| LEO | zero-shot | 100.3 | 0.00 | 17.4 | 39.1 | 42.7 | 13.3 | 0.00 | 3.00 | 5.00 | 32.3 | 0.00 | 0.00 | 7.00 | 15.3 | 59.2 |
| LEO+Spartun3D | OPT1.3B | 121.3 | 7.0 | 20.1 | 45.3 | 47.7 | 224.6 | 30.6 | 24.9 | 53.2 | 66.9 | 229.7 | 44.8 | 32.1 | 60.9 | 83.8 |
| | Vicuna7B | 125.4 | 10.1 | 22.1 | 46.7 | 52.1 | 238.9 | 32.1 | 24.4 | 55.0 | 68.3 | 242.1 | 46.5 | 35.2 | 63.1 | 84.3 |
| LEO*+Spartun3D | Vicuna7B | 129.2 | 10.4 | 23.0 | 48.1 | 53.2 | 211.3 | 32.1 | 24.6 | 55.0 | 67.8 | 247.1 | 47.5 | 36.2 | 65.1 | 85.8 |
| Spartune3D-LLM | OPT1.3B | 124.1 | 9.2 | 21.0 | 47.3 | 49.4 | 227.2 | 31.4 | 26.3 | 54.1 | 68.2 | 232.3 | 45.2 | 33.2 | 62.1 | 85.4 |
| | Vicuna7B | 131.2 | 10.3 | 24.3 | 48.8 | 53.7 | 240.4 | 32.1 | 25.0 | 55.3 | 68.7 | 244.0 | 47.1 | 36.4 | 64.0 | 86.8 |
| Spartune3D-LLM* | Vicuna7B | 135.4 | 10.7 | 24.9 | 51.3 | **56.9** | 254.7 | 32.9 | 26.7 | 57.3 | 69.7 | 252.1 | 47.6 | 36.2 | 65.4 | 88.7 |

Table 3: Experimental Results on SQA3D given the 3D objects from Mask3D and Ground-truth.

| | # | Methods | Mask3D (Schult et al., 2023) | | | | GT | | | |
|---|---|---|---|---|---|---|---|---|---|---|
| | | | C | M | R | EM | C | M | R | EM |
| Zero-shot | 1 | LEO (Huang et al., 2023) | 14.2 | 6.4 | 8.2 | 12.4 | 15.3 | 6.7 | 8.6 | 13.9 |
| | 2 | LEO+Spartun3D | 82.3 | 14.2 | 32.8 | 34.7 | 83.1 | 15.2 | 33.7 | 35.9 |
| | 3 | Spartun3D-LLM | 83.5 | 15.7 | 34.7 | 36.2 | 85.6 | 16.6 | 35.8 | 37.1 |
| Fine-tune | 4 | 3D-Vista (Zhu et al., 2023) | - | - | - | 48.5 | - | - | - | - |
| | 5 | 3D-LLM (Hong et al., 2023) | - | - | - | 50.2 | - | - | - | - |
| | 6 | LEO (Huang et al., 2023) | 132.0 | 33.0 | 49.2 | 52.4 | 132.3 | 34.3 | 51.4 | 52.5 |
| | 7 | LEO*+Spartun3D | 134.0 | 34.6 | 52.2 | 53.5 | 135.3 | 34.2 | 52.1 | 54.2 |
| | 8 | Spartun3D-LLM* | **138.2** | **35.3** | **53.4** | **54.8** | **138.3** | **35.4** | **53.7** | **55.0** |

**3D Object-Text Alignment.** Inspired by the success of 2D Visual-Language models, which effectively leverage semantically aligned text and visual features to excel in downstream tasks (Radford et al., 2021; Li et al., 2022; 2023), we aim to enhance the 3D visual representations so that they can better encode the situated spatial information and effectively align with the textual descriptions. Specifically, we introduce a 3D object-text alignment loss to guide the learning process of point cloud encoders within 3D-based LLMs, leveraging the robust language representations captured by pre-trained text encoders. We experiment with various text encoders, and CLIP achieved the best performance. For more details, please refer to the Sec. A.4 in the Appendix.

More formally, we obtain the text representations of situated textual description for each object from pre-trained text encoders, denoted as $\mathbf{W} = [\mathbf{w}_1, \mathbf{w}_2, ..., \mathbf{w}_k] \in \mathbb{R}^{K \times D}$. For object visual representations $\mathbf{O}$, we employ spatial self-attention layers (Chen et al., 2022) to learn spatial-aware object representations. Specifically, a pairwise spatial feature matrix $\mathbf{F} \in \mathbb{R}^{K \times K \times 5}$ is introduced to represent relative spatial relations between objects. For example, for each object pairs $\mathbf{o}_i$ and $\mathbf{o}_j$, we construct pairwise spatial feature as $\mathbf{f}_{ij} = [d_{ij}, sin(\theta_h), cos(\theta_h), sin(\theta_v), cos(\theta_v)] \in \mathbb{R}^{1 \times 5}$, where $d_{ij}$ is Euclidean distance between two objects, and $\theta_h$ and $\theta_v$ are horizontal and vertical angles connecting bounding box centers of $\mathbf{o}_i$ and $\mathbf{o}_j$, respectively. Then, we inject $\mathbf{F}$ into the self-attention of the object as,

$$\mathbf{O}' = \texttt{softmax}(\frac{\mathbf{Q}\mathbf{K}^T}{\sqrt{d_h}} + \texttt{MLP}(\mathbf{F}))\mathbf{V}, \text{where } Q = W_Q O; K = W_K O, V = W_V O, \quad (2)$$

where $\mathbf{O}' \in \mathbb{R}^{K \times D}$ denotes the spatial-aware object representation; Then we use a Mean Squared Error (*i.e.*, MSE) as the objective function to minimize the distance between the object representation $\mathbf{O}'$ and the corresponding situated textual embedding $\mathbf{W}$, denoted as $\mathcal{L}_{\texttt{align}} = \texttt{MSE}(\mathbf{o}', \mathbf{w}_t)$. The model is trained to jointly optimize both the alignment loss and the language modeling loss (Eq. 1) as $\mathcal{L} = \mathcal{L}_{LM} + \mathcal{L}_{\texttt{align}}$.

## 5 EXPERIMENT

### 5.1 EXPERIMENTAL SETUP

To demonstrate the effectiveness of our proposed Spartun3D-LLM, we conduct experiments on two situated understanding datasets: Spartun3D[1] and SQA3D. For SQA3D, we evaluate under two conditions: object proposals in 3D are derived either from Mask3D (Schult et al., 2023) or ground-truth annotations. Also, we assess the transferability of our method on the navigation task using MP3D ObjNav (Savva et al., 2019). Following LEO (Huang et al., 2023), we report the performance

---

[1] https://github.com/zhangyuejoslin/Spartun3D

using standard generation metrics, including *CIDEr*, *METEOR*, *BLEU-4*, and *ROUGE_L*, sentence similarity (Reimers, 2019) for captioning task. For SQA3D and situated QA tasks of questions about attributes and relations, we also report an additional metric of exact-match accuracy. More details about metrics are introduced in Appendix A.2.1. We also provide implementation details in Sec. A.3. We leverage LEO as baseline. Since the training stage in LEO has covered most of the evaluation tasks, we experiment with models initialized from scratch to ensure a fair comparison in the zero-shot setting. For other settings, we report the performance of models initialized both from scratch and from the instruction-tuned LEO. To distinguish between the two, models initialized from the instruction-tuned LEO are marked with an asterisk ($^*$).

## 5.2 EXPERIMENTAL RESULTS

**Spartun3D Benchmark.** We evaluate the performance of both the LEO model and Spartun3D-LLM after fine-tuning them on our proposed Spartun3D dataset. The fine-tuned LEO model is referred to as LEO+Spartun3D. Table 2 and Table 4 show the experimental results on captioning and QA tasks, respectively. We experiment with two different LLM backbones: Opt1.3B and Vicuna7B. Our experiments show that Spartun3D-LLM consistently outperforms LEO+Spartun3D across all question types (around $2\%-3\%$ across all metrics), regardless of the LLM backbone used, indicating the effectiveness of our explicit alignment module. We observe that initializing our model with LEO pre-trained weights improves performance. Notably, without fine-tuning, LEO performs reasonably well on attribute and relation questions in a zero-shot setting but struggles with other situated tasks.

**SQA3D Performance.** We evaluate our method on the SQA3D dataset, whose scenes are derived from ScanNet (Dai et al., 2017). Their scenes differ from those in Spartun3D, which are sourced from 3RScan. We experiment with two settings: zero-shot and fine-tuning. In the zero-shot setting, we re-trained LEO on their dataset (row#1) only constructed from 3RScan excluding all dataset constructed from ScanNet to ensure a fair comparison with our method. As shown

Table 4: Experimental Results on Spartun3D Situated Captioning Task.

| Models | LLMs | C | B-4 | M | R | S |
|---|---|---|---|---|---|---|
| LEO | zero-shot | 0.00 | 0.00 | 9.00 | 15.3 | 51.9 |
| LEO+Spartun3D | OPT1.3B | 5.9 | 15.3 | 17.7 | 31.2 | 67.3 |
| | Vicuna7B | 6.7 | 15.8 | 18.7 | 32.3 | 70.4 |
| LEO+Spartun3D* | Vicuna7B | 14.1 | 17.2 | 22.6 | 32.1 | 76.3 |
| Spartun3D-LLM | OPT1.3B | 6.4 | 15.7 | 18.5 | 31.2 | 68.6 |
| | Vicuna7B | 8.5 | 16.4 | 19.6 | 32.5 | 72.5 |
| Spartun3D-LLM* | Vicuna7B | **14.6** | **19.3** | **23.3** | **33.4** | **78.1** |

in Table 3, LEO performs poorly on SQA3D in the zero-shot setting, suggesting its limitations in learning situated understanding from its dataset. In contrast, LEO trained on Spartun3D (row#2) shows significant improvement, demonstrating the effectiveness of our dataset. Further comparisons of Spartun3D-LLM with LEO+Spartun3D demonstrate a better zero-shot learning (i.e., generalization) capability of our model. In the fine-tuning setting, Spartun3D-LLM continues to outperform LEO across all metrics. Table 8 in the Appendix provides a breakdown of the fine-tuned performance across various question types, where Spartun3D-LLM shows consistent improvement.

**Navigation Performance.** To demonstrate the effectiveness of our approach on downstream embodied tasks, we evaluate it on the object navigation tasks. Specifically, we randomly select 5 scenes that contain around 1000 examples from the MP3D ObjNav dataset. In this task, we additionally input 2D ego-centric images to both LEO and Spartun3D-LLM for comparison. There are four types of nav-

Table 5: Performance on Navigation. (Accuracy %)

| | LEO | Spartun3D-LLM |
|---|---|---|
| Zero-shot | 0 | **20.3** |

igation actions: turn left, turn right, move forward, and stop. We evaluate whether the model generates correct action at each step. We conduct the experiment in a zero-shot setting, and Table 5 shows the accuracy of the model's performance. The baseline model, LEO, struggles to generate the required action-related text to guide navigation steps without fine-tuning specifically for navigation tasks. In contrast, our model demonstrates strong transferability to generate correct actions. Fig 8 (e) showcases a qualitative example, illustrating how our model effectively generates accurate navigation actions without task-specific fine-tuning.

## 5.3 ABLATION STUDY AND EXTRA ANALYSIS

**Explicit Alignment Enhances General Spatial Understanding.** We evaluate the effectiveness of our proposed situated spatial alignment module on general scene understanding tasks, such as Scan2Cap (Chen et al., 2021) and ScanQA (Azuma et al., 2022). In line with our approach for

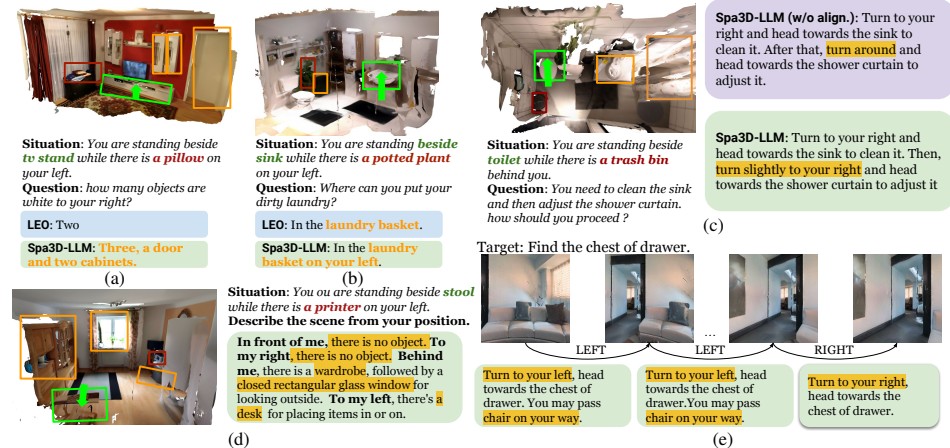

Figure 8: Qualitative Examples. (a), (c), (c) situated qa examples of object attribute and relation, affordance, and planning, respectively. (d) situated captioning. (e) navigation in a zero-shot setting. Our model generates both actions and descriptions of surrounding objects while navigating.

situated tasks, we construct textual descriptions for each object based on its attributes and spatial relations to others from a top-view perspective. As shown in Tab. 6, by incorporating the explicit spatial alignment module, our model shows better results, indicating that our proposed alignment module not only improves situated understanding but also enhances general 3D scene understanding.

**Improved Situated Understanding.** To analyze the model's situated understanding ability further, we visualize the distribution of model responses generated for questions requiring strong spatial understanding from SQA3D. Specifically, we extract questions starting with *"which direction"*. Fig. 10 illustrates the distribution of generated *"directions"*, including *"left"*, *"right"*, *"forward"* and *"backward"*. We observe that LEO is biased towards generating *"left"* 97% of the time. However, the

Table 6: Spatial Alignment Evaluation on Other Benchmarks. The metric is Sentence Similarity (%).

| Methods | Scan2Cap | ScanQA |
|---|---|---|
| LEO + Spartun3D | 54.2 | 46.3 |
| Spartun3D-LLM | **55.7** | **48.6** |

ground-truth distribution of *"left"* and *"right"* should be balanced, suggesting that LEO may have a limited understanding of situated spatial relationships. The bias is significantly mitigated when LEO is trained on our dataset (LEO+Spartun3D). While adding our alignment loss (Spartun3D-LLM) helps futher, our dataset is the primary factor in addressing the bias.

**Scaling Performance.** We conduct scaling experiments to demonstrate how model performance improves with the addition of Spartun3D datasets. As shown in Fig. 9, we evaluate performance on SQA3D and observe consistent improvement as the dataset scales, highlighting the potential for dataset expansion using our proposed method.

**Qualitative Examples.** In Fig. 8, we showcase several successful examples to demonstrate the effectiveness of Spartun3D-LLM across various situated tasks. Notably, in Fig 8 (c), the model without an explicit alignment module tends to generate more general or vague spatial descriptions, such as *"turn around"*. In contrast, with the alignment module, the model produces more specific details, including terms like *"turn slightly right"*. To verify this, we examine 30 examples from both situated planning and situated captioning tasks and observe this phenomenon in 17 of them. This highlights how the proposed spatial alignment module enhances the generation of fine-grained spatial information, leading to more precise and contextually accurate outputs.

## 6 DISCUSSION AND CONCLUSION

In this work, our goal is to address the limitation of situated understanding of the 3D-based LLMs from two perspectives. First, we propose a method to construct an LLM-generated dataset based on our designed situated scene graph. Then, we propose an explicit situated spatial alignment on the 3D-LLM to encourage the model to learn alignment between 3D object and their textual representations directly. Finally, we provide comprehensive experiments to show our own benchmark improve situated understanding of SQA3D and navigation. We also provide analysis to show our proposed explicit alignment module helps spatial understanding.

## 7 ACKNOWLEDGEMENT

This project is partially supported by the Office of Naval Research (ONR) grant N00014-23-1-2417, the award No. 2238940 from the Faculty Early Career Development Program (CAREER) of the National Science Foundation (NSF) and the U.S. DARPA FoundSci Program #HR00112490370. Any opinions, findings, and conclusions or recommendations expressed in this material are those of the authors and do not necessarily reflect the views of Office of Naval Research.

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

# A  APPENDIX SECTION

## A.1  HUMAN EVALUATION

**Human Scores.**    To evaluate the quality of Spartun3D based on human scores, we mainly consider the following two aspects.

- **Language Naturalness** evaluates the flow and syntax of the language, ensuring it reads naturally as if written by a human and adheres to common-sense knowledge in practical scenarios. Examples of different scoring levels for situations, questions, and answers are provided in Table 7. For instance, a score of 1 might be assigned to a scenario like *"Standing beside a blanket while there is a toilet in the background,"* which is uncommon in a typical setting. Similarly, questions such as *"Which is cleaner, the trash bin or the bed?"* are unlikely to be asked by humans, reflecting low language naturalness.

- **Spatial Fidelity** is critical to ensure that the generated data accurately represents the information in 3D scenes and adheres to factual details. This includes verifying the presence of objects within the scene and ensuring that spatial relationships between objects are correctly represented. Additionally, common sense knowledge must be considered, especially when unusual objects or scenarios are mentioned in the questions or answers. For example, as shown in Fig. 12, a score of 1 is assigned to the instance where *"clothes are hung on a mirror."* This error arises because the human-annotated data from 3RScan labeled the mirror's affordance as *"hanging,"* which misled the GPT model into generating an incorrect dataset.

**Error Analysis.**    We randomly sampled 50 examples from each task (200 in total) to validate the quality of our automatically generated data and manually assess the quality from the aspects of language naturalness and spatial fidelity. Language naturalness evaluates whether the generated texts are natural or written by a human, while spatial fidelity ensures that data accurately reflects the 3D scene. We observe 26 errors in total and summarize them into the following categories.

- **Semantic Errors**: The generated sentences may contain semantic mistakes. For instance, the answer *" You should go in front of you to water the potted plant."*.

- **Common Sense Violations**: The generated content may occasionally conflict with basic common sense knowledge, such as producing unusual questions and answers. For example, it might generate a question like "If I want to store items, which object is most accessible?" with the answer being "trash bin." This issue arises because the human-annotated data includes an affordance for trash bins as objects for storing items. Such annotations inadvertently influence GPT-4o to generate QA pairs that conflict with common sense knowledge.

- **Spatial Inconsistencies**: Errors in capturing or reasoning about spatial relationships in the 3D environment primarily occur in Situated Planning tasks, which demand complex two-step spatial reasoning. These errors often arise because the second action depends on the outcome of the first action, and inaccuracies sometimes occur during the generation of the second action.

- **Misalignment between visual context and textual descriptions.** In some cases, the agent's view is obstructed due to the room layout or object size. For example, consider the situation: "Standing beside the sofa, there is a closet on your left." However, the closet is actually located in another room and cannot be seen from the agent's current standpoint. To address this issue, we designed a scenario where the agent stands beside a pivot object and consistently faces the center of the pivot object, rather than facing a random object that could potentially be obstructed. Additionally, we incorporated pass-by spatial information to enhance the agent's awareness of surrounding objects, providing a more comprehensive sense of the environment.

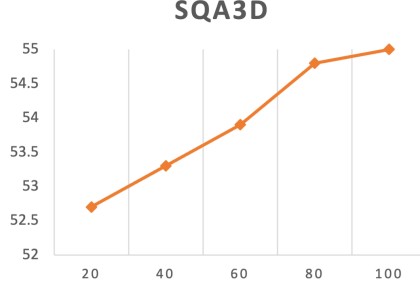

Figure 9: Scaling Effect on SQA3D.

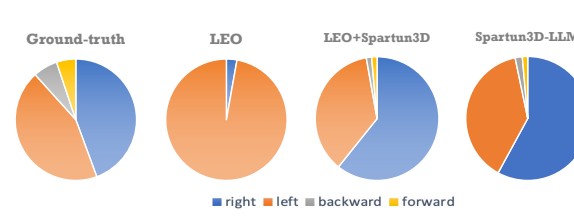

Figure 10: SQA3D Labels Distribution

| Scores | Situation | Question | Answer |
|---|---|---|---|
| 5 | You are standing beside the nightstand while there is a picture on your left. | How many sofa chairs are to my left? | You can use the lamp behind you, but be careful of the stools you will pass by. |
| 4 | You are standing beside a lamp while there is a box on your left. | Is the bag closer to me than the trash bin behind me? | You can place items on the cabinet to your left, which is the closest option. Alternatively, you can place them on the bed to your left. |
| 3 | You are standing beside the radiator while there is a curtain rod on your back. | I want to read a book. Should I walk to the window or sit in the chair? | You can go to the clothing hanging on the door behind you. |
| 2 | You are standing beside salt while there is a kitchen cabinet on your left. | What are the characteristics of the window to my left? | Turn slightly back to your right and head towards the lamp to move it closer to your bed. You may pass by the curtain rod. |
| 1 | You are standing beside a blanket while there is a toilet on your back. | Which is cleaner, the trash bin behind me or the bed to my left? | You should go in front of you to water the potted plant. |

Table 7: Examples to Evaluate Language Naturalness.

## A.2 EXPERIMENTAL RESULTS

### A.2.1 EVALUATION METRICS

- **CIDEr** (Consensus-based Image Description Evaluation): A metric that measures the similarity between generated and reference descriptions by assessing n-gram overlap, tailored for image captioning but applicable in NLP for structured comparisons.

- **METEOR** (Metric for Evaluation of Translation with Explicit ORdering): A metric that evaluates machine translation and other language generation tasks, considering precision, recall, and harmonic mean, while accounting for synonyms and stemming.

- **ROUGE-L** (Recall-Oriented Understudy for Gisting Evaluation, Longest Common Subsequence): This metric focuses on the longest common subsequence between generated and reference texts, providing insight into recall and precision in longer sequences.

- **EM**: An evaluation metric that checks whether the predicted output matches the reference exactly, often used in tasks like question answering.

## A.3 IMPLEMENTATION DETAILS.

The maximum context length and output length of LLM are both set to 256. For each 3D scene, we sample up to 60 objects with 1024 points per object. During training, the pre-trained 3D point cloud encoder and the LLM are frozen. We set rank and $\alpha$ in LoRA to be 16 and dropout rate to

|  | What | Is | How | Can | Which | Others |
|---|---|---|---|---|---|---|
| CLIPBERT (Ma et al., 2022) | 39.7 | 46.0 | 40.5 | 45.6 | 36.1 | 38.4 |
| SQA3D (Ma et al., 2022) | 31.6 | 63.8 | 46.0 | 69.5 | 43.9 | 45.3 |
| 3D-LLM (Hong et al., 2023) | 35.0 | 66.0 | 47.0 | 69.0 | **48.0** | 46.0 |
| 3D-Vista (Zhu et al., 2023) | 34.8 | 63.3 | 45.4 | **69.8** | 47.2 | 48.5 |
| LEO (Huang et al., 2023) | 46.8 | 64.1 | 47.0 | 60.8 | 44.2 | 54.3 |
| Spartun3D-LLM | **49.4** | **67.3** | **47.1** | 63.4 | 45.4 | **56.6** |

Table 8: Exact Match Performance on SQA3D Across Various Question Types

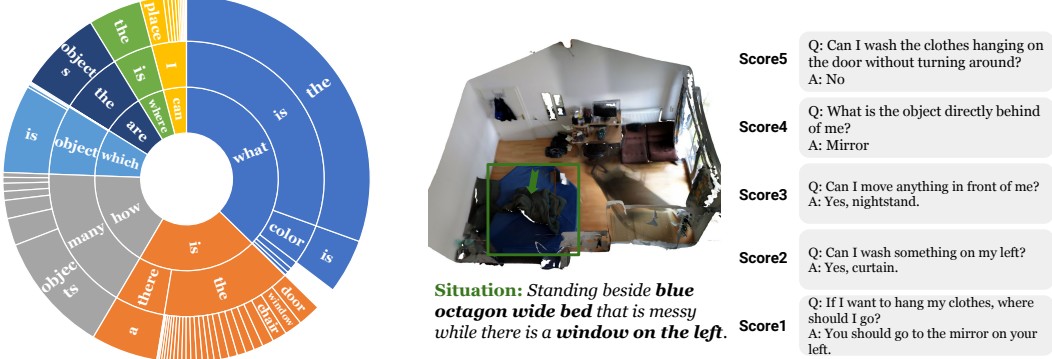

Figure 11: Spartun3D Word Distribution.

Figure 12: Examples to Evaluate Spatial Fidelity.

be 0. During inference, we employ beam search to generate the textual response, and the number of beams is 5. The model is trained on a 6 NVIDIA RTX A6000 GPU for around 30 hours with 15 epochs. The learning rate is $3e-5$, and the batch size is 24.

## A.4 EXTRA EXPERIMENTAL RESULTS

**Different Text Encoder.** We conducted an ablation study to evaluate the impact of different text encoders on enhancing 3D visual representations. Table 9 shows the results on object attribute and relation tasks. Our results indicate that text encoders from VL models outperform pure text encoders, highlighting the importance of multimodal learning for 3D object understanding. Among the tested encoders, the CLIP encoder yields the best results.

| Text Encoder | EM |
|---|---|
| BERT (Reimers, 2019) | 53.5 |
| BLIP (Li et al., 2022) | 54.2 |
| CLIP (Radford et al., 2021) | **56.9** |

Table 9: Different Text Encoder Evaluation.

Table 10: Prompts for Different Tasks.

**Prompts for Situated Captioning:** Provide a summary of a scene focusing on object types and attributes ON MY LEFT/RIGHT/FRONT/BACK. Describe the scene also considering common sense, such as how objects can be used by humans and human activities in the left part of the scene.The description should conform to the given scene information. You don't need to describe each object in the scene, pick some objects of the scene for summary. You can also summarize the room's function, style, and comfort level based on the arrangement and color of objects within the room. Your summary must be one paragraph, not exceeding 300 words. Don't use IDs of the objects in the summary. Don't use turn degrees or distance meters in the summary.

**Prompts for Object Attributes and Relations:** Ask questions about object types, and counting. The questions related to attributes are better be asked when multiple objects contain the same attributes and the answer can be specified based on spatial relations. You can also ask about spatial positions between me and other objects or spatial relations between objects by comparing the angles. Based on 'relations' in the scene graph, you can ask about other relations between objects.

**Prompts for Object Affordance:** Ask questions about object affordance and object utility based on common sense. The answer should consider the best option that follows common sense knowledge and is closer to me. If I plan to go to some objects, other objects are blocking my way; please specify them. There are several examples:
Q: Where should I go to quickly put something down?A: You can use the chair in front of you.
Q: I want to read a book. Should I walk to the window or sit in the chair? A: Sit in the chair, which is closer.
Q: If I want to reach the kitchen counter, what object will be passed by? Answer: tables with chairs.
Q: What should I do if I want to cook? A: You can go to the kitchen area, but be careful about the tables and chairs you will pass by.

**Prompts for Situated Planning:** You need to generate 6 meaningful question-answer pairs that require multi-hop reasoning and planning based on the scene information.Ask questions about object affordance and object utility based on common sense and path planning. The question must be answered based on my position. If I plan to go to some objects, other objects are blocking my way, please specify them. The turn action should be considered based on angles if I plan to go to multiple places. Do not use the number of turn degrees or distance meters in the question and answer. There are several examples:
1. Question: I want to dim the lights and take a nap; What should I do? Answer: Turn to your right and head towards the lamp. Dim the lights, then turn slightly to your left and head towards the sofa to lie down.
2. Question: I want to light up the area near the kitchen counter to prepre some food. How should I proceed? Answer: Turn slightly to your left and head towards the blinds on your left to adjust them. Then, turn slightly back to your right and head towards the kitchen cabinet in front of you.
3. Question: I need to adjust the lighting to make the room brighter. What should I do? Answer: Turn to your left and head towards the lamp. Adjust the lighting. Then, turn slightly back to your right and ensure the curtains or blinds are open to let in more light.
4. Question: I need to adjust the lighting to make the room brighter and then prepare a snack on the kitchen counter. How should I proceed? Answer: Turn to your left and head towards the lamp to adjust the lighting. After adjusting the lighting, turn slightly back to your right and head towards the kitchen counter. You may pass tables and chairs on your way.

Table 11: Prompts used to Generate Situated QA related to Object Attribute and Relation.

**General Prompts for Object Attribute and Relation:** You need to generate at least 10 meaningful question-answer pairs based on the scene information. Ask questions about object types, and counting. The questions related to attributes are better be asked when multiple objects contain the same attributes and the answer can be specified based on spatial relations. You can also ask about spatial positions between me and other objects or spatial relations between objects by comparing the angles. Based on 'relations' in the scene graph, you can ask about other relations between objects. You need to provide the queried object. Do not consider the object's utility and affordance. Do not use the number of turn degrees or distance meters in the question and answer. Do not use the IDs of the objects in the question and answer. The question-answer pair should be following format: Q: <question>T: <queried object_id(s)>A: <Answer>. You can answer the question according to the queried object(s). If there is no information about the question, the <Answer>should be "unkown". There are several examples:
Q: What is the object closest to the left of me? T:lamp_1 Answer: a lamp.
Q: How many stools are on my left? T:stool_3 A: One.
Q: There are multiple chairs, what is the size of the chair left of me? T:chair_1, chair_2 A: low chair.
Q: Is the cabinet far from me or the sofa far from me? A: sofa.
Q: How many black objects are to my right? A: Two, a towel and a toilet brush.
Q: Where is the trash bin? A: Behind you.
Q: What color is the trash bin in front of me? A: black (one white left of me and one black in front of me)
Q: Is the mirror right of the shower curtain based on my standing position? A: Yes.
Q: Is the light on my right on or off?
Q: What is the object to the left of the white heater to my right?
Q: Is there a picture to my right?
Q: Is the door in front of me the same color as the cabinet to my right?
Q: The tv to your 11 o ' clock direction on; true or false ?
Q: Can black objects are to my right be divided by three?

| **Coordinate prompt:** | **Spatial Prompt:** |
|---|---|
| You are standing beside the white toilet_6, and the initial 3d coordinate is [-1.15, 0.29, 0.48] toilet_6. You are facing the center of the toilet_6, and the center coordinate of toilet_6 is [-1.36, 0.28, 0.48]. The scene contains some objects, which compose a scene graph in JSON format describing objects, such as object coordinate, color, size, shape, and state. You can calculate object distance and rotation angle related to your standing point and orientation using coordinates. If the rotation angle is in the range [315-360,0-45] is defined as Front, [45-135] is RIGHT, [135-225] is BACK, and [225-315] is right. For example, from the scene graph `"table_8": "coordinate": [1,1,1], "affordances": ["placing items on"], "attributes": "color":"red", "relations": ["close by chair_36"].` We know that the coordinate of table_8 is [1,1,1], there is a table_8 that is close by chair_36. You can place items on this table_8. | You are standing beside a white toilet_6. The scene contains some objects, which compose a scene graph in JSON format with four keys: "left", "right", "front", "backwards", indicating objects in the corresponding direction. Each entity in the scene graph denotes an object instance with a class label and an object ID. The "distance" indicates the meters between you and the object. The "angle" represents the degrees compared to your current direction, where your direction in front is 0 degrees. The larger angles are means further right. The "affordance" is the motion activity related to this object. The "attributes" describe the object's characteristics, such as 'color' and "size". The "relations" describe the spatial relationships with other objects. The "passby" indicates other objects in your path if you walk toward it from your current position. For example, from the scene graph `"Left":"table_8": "distance": 2.6, "passby": ["chair_21"], "affordances": ["placing items on"], "attributes": "color":"red", "angle": 257.48, "relations": ["close by chair_36"].` We know that on my left 257.48 degrees and 2.6 meters, there is a table_8 that is close by chair_36. You can place items on table_8. If you go to table_8, you could pass by chair_21. |

