# OpenReview forum: "SPARTUN3D: Situated Spatial Understanding of 3D World in Large Language Model"
_ICLR.cc/2025/Conference — ICLR 2025 Poster_

### Official Review · Reviewer_kEmt · 2024-10-30

**Soundness:** 2
**Presentation:** 3
**Contribution:** 2
**Rating:** 6
**Confidence:** 4

**Summary:**

SPARTUN3D addresses key limitations in current 3D LLMs:

- Existing 3D datasets are lack of situated context
- Current 3D LLM's architecture lack explicit alignment between the spatial representations of 3D scenes and natural language.

To tackle these issues, this paper introduces a novel dataset, SPARTUN3D, along with a model enhancement named SPARTUN3D-LLM, featuring a dedicated spatial alignment module. Experimental results validate that both the dataset and model improvements significantly advance the situated spatial reasoning capabilities of 3D LLMs.

**Strengths:**

(1). This paper is very well-motivated. The importance of situational reasoning in 3D-LLM is beyond doubt.

(2). The writing of the paper is sound.

(3). This paper covers a range of tasks that are less explored by 3D-LLMs.

**Weaknesses:**

(1). The motivation behind this dataset is strong, and I acknowledge the considerable effort put into curating it. However, the dataset’s quality remains unverified. While the authors took steps to avoid unnatural configurations (see line 176), I am not yet convinced of its robustness. Specifically, the input scene graph includes angular relationships calculated using the centroids of three objects. Given the variability in object sizes, it is uncertain whether GPT-4o can accurately interpret and annotate these spatial relations. What evidence supports the notion that these spatial relationships are meaningful to GPT-4o, and do these annotations align with human perception?

(2). GPT-4o’s capacity to generate large volumes of data makes it ideal for annotating 3D scenes at scale. However, the paper lacks scaling experiments to demonstrate how model performance improves as more data is generated using this pipeline. Including such experiments would provide valuable insight into the benefits of scaling for this approach.

(3). The input to the model is triple <C, S, Q>, where C is the 3D scene context, S is the situation, and Q is a question. The situation S can be further denoted as $S^t$, $S^p$, $S^r$, where $S^p$ is the 3D coordinate in the form <x, y, z> and $S^r$ is the quaternion. Can author provide more analysis on how <x, y, z> and $S^r$ would affect models' performance?

(4). The setting is egocentric situational 3D-LLM, but SPARTUN3D takes mostly text information as input. Is it possible to directly takes a 2D image as egocentric condition?

(5). For the experiments Navigation Performance in zero-shot manner, LEO's training data is very simple. It will be interesting to see a comparison between finetuned LEO with SPARTUN3D and Sparttun3D-LLM.

**Questions:**

(1). The input to the model is triple <C, S, Q>, where C is the 3D scene context, S is the situation, and Q is a question. The situation S can be further denoted as $S^t$, $S^p$, $S^r$, where $S^p$ is the 3D coordinate in the form <x, y, z> and $S^r$ is the quaternion. In the dataset, line 182, authors assume the agent's orientation is always facing forward to the center of the selected object. How is this selected object chosen? Will this introduce any bias?

(2). Can author show how they make inference of SPARTUN3D? More specifically, what is S, C, and Q respectively? Is it possible to provide an example?

(3). Why does finetuned 3D-LLM only has EM score? I understand 3D-Vista is a BERT but 3D-LLM seems not?

---

> ### Author Response · Authors · 2024-11-23
>
> We sincerely appreciate the reviewer’s acknowledgment of our motivation and contribution, and we mainly address the reviewer's comments and questions below.
>
> **1. Quality Control.**
>
> a) **What evidence supports the notion that these spatial relationships are meaningful to GPT-4o?**  Thank you for the questions. In fact, based on our experiment in Sec.3.4, GPT-4o's ability to interpret spatial relations is strongly related to different prompts.  We experiment with two types of prompts for representing spatial information to prompt GPT-4o: **Cord-prompt**, which consists of  object center coordinates, standing point, orientation, and instructions for calculating distances and rotation angles,  and **Spa-prompt**, consisting of the calculated angles and distance based on the approaches we described in Sec.3.3. An example of each type of prompt can be found in Tab.11 in the Appendix. The following table indicates that Cord-prompt yields unsatisfactory results, revealing that LLMs lack strong 3D spatial reasoning capabilities when interpreting raw spatial coordinates. Our Spa-prompt significantly improves the quality of the generated dataset by providing qualitative spatial relations (e.g. distance, direction). **The role of GPT-4o in our approach is primarily to organize this pre-processed spatial information and generate QA data. Therefore, there is minimal reliance on GPT-4o for interpreting the 3D scene environment, as nearly all spatial information is already systematically structured.**
>
> |                | Captioning | Atti and Rel. | Affordance | Planning |
> |----------------|------------|---------------|------------|----------|
> | **Cord-Prompt** | 0.27       | 0.57          | 0.4        | 0.33     |
> | **Spa-Prompt**  | 0.86       | 0.9           | 0.87       | 0.87     |
>
>
> Besides, we would like to provide an example of how we prompt GPT-4 to help the reviewer gain a clearer understanding of our method. When prompting GPT-4o, we provide an explanation of how to interpret the spatial information. GPT-4o demonstrates a strong understanding of this pre-processed spatial information. Below is an example, which is also presented in Table 10 of the Appendix.
>
> ``{"Left":{"table\_8": {"distance": 2.6, "passby": ["chair\_21"], "affordances": ["placing items on"], "attributes": {{"color":"red"}}, "angle": 257.48, "relations": ["close by chair\_36"]}}}.``
>
> ``From this situated scene graph, we know that on my left 257.48 degrees and 2.6 meters, there is a table\_8 that is close by chair\_36. You can place items on table\_8. If you go to table\_8, you could pass by chair\_21.``

---

> ### Author Response · Authors · 2024-11-23
>
> b) **How this spatial information aligns with human perception.** Thank you for the question. To evaluate whether GPT-4o is able to interpreate the spatial information and generate the correct dataset, we conduct a comprehensive human evaluation and introduce human scores based on two key criteria: *language naturalness*, which evaluates whether the text reads as if it were naturally written by a human, and *spatial fidelity*, which ensures that the data accurately reflects the 3D scene with correct spatial relationships. The detailed explanations are as follows.
>
> - **Language Naturalness**: evaluates the flow, syntax, and semantics of the language, ensuring it reads naturally as if written by a human and adheres to common-sense knowledge in practical scenarios. For instance, a score of 1 could be assigned to a scenario like ``Standing beside a blanket while there is a toilet in the background,`` which is uncommon in a typical setting. Similarly, questions such as ``Which is cleaner, the trash bin or the bed?`` are unlikely to be asked by humans, reflecting low language naturalness.
>
> - **Spatial Fidelity**: is critical to ensure that the generated data accurately represents the information in 3D scenes and adheres to factual details. This includes verifying the presence of objects within the scene and ensuring that spatial relationships between objects are correctly represented. Additionally, common sense knowledge must be considered, especially when unusual objects or scenarios are mentioned in the questions or answers. For example, in a 3D scene, a score of 1 is assigned to the instance where ``clothes are hung on a mirror.`` This error arises because the human-annotated data from 3RScan labeled the mirror's affordance as ``hanging,`` which misled the GPT model into generating an incorrect dataset.
>
> Each criterion is rated on a scale from $1$ to $5$, and the average of these two scores is the overall human score. We randomly select $50$ examples from each task and compute human scores of situation, question, and answer, respectively. We mainly have the following findings:
>
> 1) The average scores align with the complexity of each task, with relatively lower scores for captioning and planning tasks.
>
> |                Task                | Situation | Question | Answer |
> |------------------------------------|-----------|----------|--------|
> | **Captioning**                     | 4.81      | -       | 4.17     |
> | **Attr and Rel.** (Attributes/Relations) | 4.74      | 4.63     | 4.56   |
> | **Affordance**                     | 4.72      | 4.51     | 4.47   |
> | **Planning**                       | 4.75      | 4.22     | 4.05   |
>
> 2) To assess how our generated data compares to human-annotated data, we sampled $50$ examples from SQA3D and mix them with our dataset. We focus on the human score of different types of questions. As shown in the following table, our generated questions are comparable to the questions in SQA3D across various question types.
>
> |            | What  | Is    | How   | Can   | Which | Other  |
> |------------|-------|-------|-------|-------|-------|--------|
> | **SQA3D**  | 4.81  | 4.71  | 4.41  | 4.52  | 4.65  | 4.73   |
> | **Spartun3D** | 4.62  | 4.57  | 4.31  | 4.45  | 4.51  | 4.64   |

---

> > ### Author Response · Authors · 2024-11-23
> >
> > **2. Scaling Curve.** Thank you for pointing this out. Reviewer jrTE also highlighted this important issue. To address it, we have included a scaling curve in the updated version. We conducted scaling experiments to evaluate our model's performance on SQA3D, and the results demonstrate consistent improvements as Spartun3D scales up.
> >
> > | % of Dataset | EM   |
> > |--------------|-------|
> > | 20           | 52.7  |
> > | 40           | 53.3  |
> > | 60           | 53.9  |
> > | 80           | 54.7  |
> > | 100          | 55.0  |
> >
> >
> > **3. More analysis on how standing points and orientation influence the results.** Thank you for your feedback! As stated in our paper (Lines 322–333), during training, both  S^p  (standing points) and S^r  (orientations) are provided to enable environment rotation and translation. This setup allows the model to learn spatial transformations effectively. However, during testing, only textual situational descriptions are provided, ensuring that the model generalizes without relying on S^p  or S^r .
> >
> > To address the reviewer's concern, we conducted an additional ablation study where the model was trained without S^p  and S^r. The results are summarized in the table below. As can be seen, performance slightly decreases when these components are excluded from the training process. This indicates the robustness of the model in that the model could learn situations from text, not only relying on standing points and orientations.
> >
> > |                        | Attri. and Rel. | Affordance | Planning |
> > |------------------------|-----------------|------------|----------|
> > | **Spartun3D-LLM**      | 56.9           | 69.7       | 88.7     |
> > | **w/o S^p and S^r**      | 55.4           | 68.8       | 86.9     |
> >
> >
> > **4.Takes 2D image as egocentric.** Exploring egocentric 2D images as situational context is indeed an interesting and promising direction. Our model is capable of taking 2D images as additional input tokens if necessary.
> > However, our work primarily focuses on grounding textual descriptions of situations directly in 3D space. In our setup, egocentric situations can be effectively represented through textual descriptions by identifying the standing point and orientation. While incorporating egocentric 2D images is an intriguing possibility, we believe our text-based situated design is both valid and sufficient for the scope of this study.
> >
> > **5. Fine-tuning on ObjNav.** We appreciate the reviewer’s interest in the fine-tuned results for ObjNav. ObjNav is a very large dataset, comprising approximately three million examples. Due to the time and computational constraints, we sampled around 60,000 examples to fine-tune Spartun3D and LEO and conducted evaluations on the validation set . The table below presents the results, which demonstrate that, based on the current training dataset, the situated reasoning capability in Spartun3D enhances navigation performance.
> >
> > |     %Acc          | LEO   | Spartun3D-LLM |
> > |---------------|-------|---------------|
> > | Fine-tune     | 49.3  | 52.4          |
> >
> >
> > **6. Why always facing to the center of the object.** Thank you for the question. The rationale behind this design is to minimize misalignment errors, as described earlier. The scenario is constructed based on templates, such as ``standing beside place A with B on the {left/right/front/back}.`` However, in some cases, place B may not be visible from the current viewpoint. To address this issue, we always make the original orientation face the pivot object. Besides, our analysis shows that if place B is chosen randomly, approximately one-third of the examples result in the agent's view being obstructed by other objects. To ensure both the accuracy and realism of our dataset, we adopted this strategy to determine the initial orientation.
> >
> > **7.Examples to make inference of SPARTUN3D?** The reviewer can refer to Figure 2 showing tasks of situated captioning and situated QA. In this example, the situation is ``standing beside blue octagon wide bed that is messy while there is a window on your left ``.
> >
> > For the situated captioning task, the question is ``describe the scene from your current position``, and answer is `` In front, there's a rectangular box and a big picture that can be hung or moved. To the right, there's a tall pillow close to another pillow. On the left, there's an artificial lamp and a desk close to a sofa chair and trash bin. ``
> >
> > For situated QA task, question is ``where is the desk located?`` and answer is ``behind you``.
> >
> > **8.Why 3D-LLM only has EM Scores.** We assume the reviewer is talking about Table 3. It is true that 3D-LLM is a generative model, but they do not report other generation metrics while fine-tuning with SQA3D. We use their released checkpoints to obtain the corresponding value: CIDEr: $127.3$ METER $30.8$, and ROUGE-L $47.9$.

---

> > > ### Comment · Reviewer_kEmt · 2024-11-26
> > >
> > > Thank you to the authors for their timely response. I appreciate the detailed comparison between prompting methods and the efforts put into this work. My concerns are largely addressed. As a result, I have decided to raise my score.

---

### Official Review · Reviewer_D9vh · 2024-11-01

**Soundness:** 3
**Presentation:** 3
**Contribution:** 3
**Rating:** 5
**Confidence:** 5

**Summary:**

This paper introduces SPARTUN3D, a dataset aimed at enhancing the spatial reasoning abilities of 3D-based large language models (LLMs) by providing situated context. The dataset includes tasks like situated captioning and QA, challenging models to respond based on the agent's dynamic perspective. Additionally, the authors propose a novel alignment module in Spartun3D-LLM, which improves alignment between 3D scene representations and textual descriptions. The paper demonstrates the model’s generalization on multiple 3D tasks, outperforming baselines in spatial understanding and navigation accuracy.

**Strengths:**

1) Innovative Dataset: SPARTUN3D addresses a key limitation in 3D-based LLMs by providing an extensive dataset that incorporates situated reasoning. This dataset, generated by leveraging GPT-4o, is both scalable and diverse, which is essential for training models in 3D scene understanding from dynamic perspectives. This is a well-motivated and relevant contribution to the field.

2) Effective Alignment Module: The Situated Spatial Alignment Module enhances model performance by aligning 3D visual representations with their textual counterparts. This module appears to be a critical contribution as it enables more accurate spatial understanding and has implications for embodied tasks. The use of spatial self-attention and a 3D object-text alignment loss to improve object-text coherence is particularly novel.

3) Comprehensive Experiments: The experimental evaluation is thorough, covering both in-domain tasks (Spartun3D and SQA3D) and out-of-domain tasks (MP3D ObjNav), with significant improvements shown in zero-shot navigation accuracy. Ablation studies and visualizations further confirm the efficacy of the alignment module in improving spatial reasoning capabilities.

4) Practical Applications: Situated spatial understanding is valuable for real-world applications, such as navigation and human-robot interaction. This paper makes a strong case for the applicability of SPARTUN3D in these areas.

**Weaknesses:**

1) The SPARTUN3D dataset is automatically generated with GPT-4o, which, while scalable, may lack the nuanced spatial and contextual fidelity found in human-annotated data. This synthetic nature raises questions about the dataset’s generalizability and its ability to represent real-world scenarios accurately.

2) The evaluation primarily focuses on situated QA, captioning, and navigation tasks, which, although useful, may not fully capture the complexity of embodied tasks (e.g., robotic manipulation or multi-step planning). Expanding the evaluation to include a wider variety of tasks would provide a more comprehensive assessment of the model's situated understanding abilities.

3) While the Situated Spatial Alignment Module is an interesting addition, the paper lacks a rigorous theoretical foundation for its design. Details on why specific techniques (e.g., spatial self-attention and MSE for alignment) were chosen and how they uniquely contribute to spatial alignment are not thoroughly explained, which could weaken the perception of this module’s novelty. Also, the added complexity of the alignment module may lead to increased computational demands, yet the paper does not discuss or benchmark these costs. For practical deployment in real-time applications or on resource-limited systems, it is essential to understand the trade-offs between the module’s benefits and its computational impact.

4) While comparisons to the LEO (2023.11 released) baseline and other similar models are present, the paper lacks a broader comparison with recent advances in spatial reasoning, navigation models or 3D tasks.

**Questions:**

1) By using GPT4 API, how to ensure the quality and accuracy of datasets? Can you provide more information on quality control measures for SPARTUN3D? Specifically, how are errors from the automated pipeline identified and handled?
2) How does the Situated Spatial Alignment Module impact computational resources, particularly in training time and memory usage? Would the module be feasible for real-time applications?
3) Have you considered evaluating Spartun3D-LLM on other embodied tasks beyond QA and navigation, such as robotic manipulation, multi-step reasoning in dynamically changing environments or some other 3D tasks like visual grounding, 3D object detection or dense caption?
4) Your chosen baselines like Leo,3D-Vista and 3D-LLM, were relesed in 2023,  the paper lacks a broader comparison with recent advances, please show some comparisons of other advances of 3D-LLMs with SPARTUN3D-LLM?
5) As the alignment is one of your contributions of this paepr. Are there other alignment strategies that were considered or experimented with?

---

> ### Author Response · Authors · 2024-11-23
> **Response to Reviewer D9vh**
>
> We sincerely appreciate the reviewer’s thorough reviews, as well as the recognition of our work's strengths and contributions from multiple perspectives. We have addressed the reviewer's concerns as follows.
>
> **1. Quality Control of Dataset** We truly appreciate the reviewer raising this important issue. In the original version of the paper, although we provided human evaluation in Sec.3.4 on how different prompting strategies influence the quality of data, we agree with the reviewer's constructive suggestions to provide nuanced spatial and contextual fidelity analysis. Therefore, we conduct a comprehensive human evaluation and introduce human scores based on two key criteria: *language naturalness*, which evaluates whether the text reads as if it were naturally written by a human, and *spatial fidelity*, which ensures that the data accurately reflects the 3D scene with correct spatial relationships. The detailed explanations are as follows.
>
> - **Language Naturalness**: evaluates the flow, syntax, and semantics of the language, ensuring it reads naturally as if written by a human and adheres to common-sense knowledge in practical scenarios. For instance, a score of 1 could be assigned to a scenario like ``Standing beside a blanket while there is a toilet in the background,`` which is uncommon in a typical setting. Similarly, questions such as ``Which is cleaner, the trash bin or the bed?`` are unlikely to be asked by humans, reflecting low language naturalness.
>
> - **Spatial Fidelity**: is critical to ensure that the generated data accurately represents the information in 3D scenes and adheres to factual details. This includes verifying the presence of objects within the scene and ensuring that spatial relationships between objects are correctly represented. Additionally, common sense knowledge must be considered, especially when unusual objects or scenarios are mentioned in the questions or answers. For example, in a 3D scene, a score of 1 is assigned to the instance where ``clothes are hung on a mirror.`` This error arises because the human-annotated data from 3RScan labeled the mirror's affordance as ``hanging,`` which misled the GPT model into generating an incorrect dataset.
>
> Each criterion is rated on a scale from $1$ to $5$, and the average of these two scores is the overall human score. We randomly select $50$ examples from each task and compute human scores of situation, question, and answer, respectively. We mainly have the following findings:
>
> 1) The average scores align with the complexity of each task, with relatively lower scores for captioning and planning tasks.
>
> |                Task                | Situation | Question | Answer |
> |------------------------------------|-----------|----------|--------|
> | **Captioning**                     | 4.81      | -        | 4.17     |
> | **Attr and Rel.** (Attributes/Relations) | 4.74      | 4.63     | 4.56   |
> | **Affordance**                     | 4.72      | 4.51     | 4.47   |
> | **Planning**                       | 4.75      | 4.22     | 4.05   |
>
> 2) To assess how our generated data compares to human-annotated data, we sampled $50$ examples from SQA3D and mix them with our dataset. We focus on the human score of different types of questions. As shown in the following table, our generated questions are comparable to the questions in SQA3D across various question types.
>
> |            | What  | Is    | How   | Can   | Which | Other  |
> |------------|-------|-------|-------|-------|-------|--------|
> | **SQA3D**  | 4.81  | 4.71  | 4.41  | 4.52  | 4.65  | 4.73   |
> | **Spartun3D** | 4.62  | 4.57  | 4.31  | 4.45  | 4.51  | 4.64   |
>
> 3) We also evaluate how different prompting strategies influence the quality of the data.  We experiment with two types of prompts for representing spatial information to prompt GPT-4o: **Cord-prompt**, which consists of object center coordinates, standing point, orientation, and instructions for calculating distances and rotation angles, and **Spa-prompt**, consisting of the calculated angles and distance based on the approaches we described in Sec.3.3. An example of each type of prompt can be found in Tab.11 in the Appendix.
> The following table shows the percentage of examples with high human scores (>=4) for each prompt across tasks. The results indicate that Cord-prompt yields unsatisfactory results, revealing that LLMs lack strong 3D spatial reasoning capabilities when interpreting raw spatial coordinates. Our Spa-prompt significantly improves the quality of the generated dataset by providing qualitative spatial relations (e.g. distance, direction).
>
> |                | Captioning | Atti and Rel. | Affordance | Planning |
> |----------------|------------|---------------|------------|----------|
> | **Cord-Prompt** | 0.27       | 0.57          | 0.4        | 0.33     |
> | **Spa-Prompt**  | 0.86       | 0.9           | 0.87       | 0.87     |
>
>
> We hope our extra analysis helps address the reviewer's concern.

---

> > ### Author Response · Authors · 2024-11-23
> >
> > **2. Other Embodied Tasks.** We appreciate the reviewer's interest in evaluating our approach to additional embodied tasks. To address this, we conducted experiments on CLIPort robotic manipulation. The dataset size is substantial; however, due to time and computational constraints, we fine-tuned our model on a sampled subset consisting of approximately 30k examples (1/10 of the full dataset) and reported the accuracy on the validation set. The input for this task includes natural language instructions, egocentric 2D observations, and object-centric 3D information. The action poses are fully discretized into 516 tokens, comprising 320 tokens for x-axis pose bins, 160 tokens for y-axis pose bins, and 36 tokens for z-rotation bins.
> >
> > The results for two manipulation tasks are presented in the following table. For comparison, we also fine-tuned LEO on the same dataset. We evaluate on two tasks: put-block-in-bowl and packing-google-objects. Our model demonstrates strong performance in manipulation tasks, showcasing its effectiveness compared to LEO.
> >
> > |                          | put-block-in-bowl | packing google objects |
> > |--------------------------|-------------------|-------------------------|
> > | **LEO**                 | 41.7             | 50.0                   |
> > | **Spartun3D-LLM**       | 47.6             | 52.3                   |
> >
> > Additionally, as requested by the reviewer, we have evaluated our approach on the dense captioning benchmark Scan2Cap, with the results presented in Table 6. Our method demonstrates improved performance compared to the baseline model.
> >
> >
> > **3. Question Related to Spatial Alignment Module.** Thank you for the question. Our proposed alignment module is inspired by the success of 2D visual-language models, which effectively align the semantics of text and visual modalities. Our motivation is very straightforward, i.e., to narrow the gap between 3D visual representations and textual representation.
> >
> > (a) **Why spatial self-attention?** The textual descriptions contain spatial relationships between objects, which should also be captured in the representations of the 3D world. Therefore, we construct pairwise spatial features between objects (distance, rotation angles) and inject such features into the self-attention of the objects.
> >
> > (b) **Spatial Alignment Strategy.** We achieve situated spacial alignment based on two key designs. First, we employ various situated tasks, such as situated captioning and QA, to help the model learn the alignment. Second, we introduce an explicit alignment module designed to directly reduce the distance between an object’s 3D visual representation and its corresponding textual representation.
> > We chose MSE as the loss function to bridge the gap between the representations of two modalities because it provides a straightforward and computationally efficient solution. While other strategies, such as constructing negative examples and employing contrastive loss, could be explored, they would significantly increase computational cost. Exploring contrastive learning is an interesting direction for future work.
> >
> > (c) **Computation Cost.** Thank you for pointing this out! The alignment module functions as an additional loss designed to improve alignment, and its computation occurs only during the training phase. When training with 6 GPUs, the total training time with aligned spatial modules increased by around one hour. Therefore, the training cost is slightly increased, and inference efficiency remains unaffected.
> >
> >
> > **4. LEO as the main baseline.** We appreciate the reviewer for raising this issue. We chose LEO for our study because it is open-source and demonstrates SOTA  performance across various tasks. We have also explored other recent 3D-based LLMs, such as Scene-LLM[1] (no available open-source code) and Chat-Scene[2]. First, the backbones of these models are quite similar, as they use 3D features and textual descriptions as input tokens for LLMs. Second, their improvements largely stem from incorporating additional 2D image inputs. When considering only 3D point clouds and textual inputs, their performance is comparable to LEO. Since our primary focus is on enhancing alignments between 3D and text, we chose LEO as a representative baseline.
> > That said, our dataset can also be applied to these models using extra 2D image inputs to evaluate potential improvements, which could be explored in future work.
> >
> > [1] Fu, Rao, et al. "Scene-llm: Extending language model for 3d visual understanding and reasoning."
> >
> > [2] Huang, Haifeng, et al. "Chat-scene: Bridging 3d scene and large language models with object identifiers."

---

> ### Author Response · Authors · 2024-11-29
>
> Dear reviewer D9vh,
>
> We sincerely appreciate the time and effort you’ve devoted to reviewing our work. We understand that your schedule may be quite busy, and we are truly grateful for your valuable feedback. As the discussion phase is nearing its end, we would greatly appreciate the opportunity to address any concerns or questions you may have. Thank you for your attention and consideration.
>
> With regard,
>
> Authors

---

### Official Review · Reviewer_jrTE · 2024-11-04

**Soundness:** 3
**Presentation:** 3
**Contribution:** 2
**Rating:** 6
**Confidence:** 4

**Summary:**

The paper addresses the task of situated reasoning where an agent is located at a certain location in the 3D space and needs to reason about the situation (caption or answer questions) from it's spatial location. The paper generates a new dataset called Spartan3D, relying on GPT-4o that lets them scale the size of the data; and proposes an explicit spatial alignment module where the object tokens are aligned to a desciption of spatial situations/relations of the corresponding object obtained via a template. The method is tested on situation question answering benchmark of SQA3D and the newly proposed Spartan3D dataset. The results show that the explicit spatial alignment module helps in question answering as well as other captioning tasks; and the additional Spartan3D dataset helps performance on all datasets.

**Strengths:**

- The paper is very well written and clearly lays down the problem and the proposed solution (with informative figures)
- The Spartan3D dataset will be useful to the community
- The proposed method obtains better results than prior state of the art on SQA3D and the proposed explicit alignment loss helps performance in all tasks (by about 1-2%)

**Weaknesses:**

- A big claim of the paper is that SQA3D is human collected dataset; and the proposed pipeline to generate SPARTUN3D can be very useful to scale up the dataset size. However, it is unclear how effective this automatically generated dataset is in comparison to human generated dataset. Specifically, the current experiments of Table-3 show that on real-world SQA3D benchmarks, using additional Spartan3D dataset on top of human collected SQA3D dataset helps performance by about 1-2% despite much larger dataset size. Additionally, just training on Spartan3D dataset results in significantly worse performance on SQA3D (by about 20%). Moreover, it is unclear if the automated data shows strong scaling behaviors -- which, if true, would push the community to scale up the datasets in a similar way that this paper proposed instead of collecting more human annotations.

 - L460-465 "In the zeroshot setting, we re-trained LEO exclusively on a dataset constructed from 3RScan to ensure a fair
comparison with our method. As shown in Table 3, LEO performs poorly on SQA3D in the zeroshot setting, suggesting its limitations in learning situated understanding from its original dataset. In contrast, LEO trained on Spartun3D shows significant improvement, demonstrating the effectiveness of our dataset and the generalization ability of our model."

The above set of lines are confusing to me.
- What exactly is this dataset constructed from 3RScan which is constructed for the zero-shot LEO baseline? Without this information, currently the conclusion seems to be that Spartan3D dataset is better than some other way of constructing a training dataset.
- I did not follow how obtaining good performance from LEO trained on Spartun3D leads to the "generalization ability of our model" conclusion? Table-3 does not show how the proposed model works with the "newly constructed 3RScan data" instead of the Spartan3D dataset.

- For navigation experiments in Table-5. what is Spartan3D-LLM trained on? Is LEO and Spartan3D-LLM identical except the explicit spatial alignment module for this experiment? I am trying to understand the main reason why LEO does not work at all for navigation while Spartan3D-LLM show some non-zero performance.

**Questions:**

- Some additional discussion / proof on why the additional automated data is useful. Scaling curves with varying amount of Spartan3D-LLM data used in training would help -- on real world benchmarks like SQA3D.

- Clarification on the newly constructed dataset from 3RScan for zero-shot LEO baselines, and generalization capabilities of the proposed model

- More details on the navigation experiments in Table-5; specifically regarding the training datasets used for the proposed model and the baselines.

---

> ### Author Response · Authors · 2024-11-21
> **Response to Reviewer jrTE**
>
> We sincerely appreciate the reviewer's acknowledgment of our strengths and contributions. We address the reviewer's comments as follows:
>
> 1) **How effective of automatically generated data compared to Human Data?** Thank you for the insightful question. It is important to emphasize that an exact match score alone does not necessarily reflect a stronger situated understanding ability. After analyzing answers generated by Spartun3D-LLM, we observe that our method significantly influences LEO's behavior.
> Please see our experiments in Section 5.3 (Improved Situated Understanding). We extract questions in SQA3D starting with *''which direction"*, and answer includes *''left''*, *''right''*, *''forward''* and *''backward''*. We observe that LEO is biased towards generating *''left''* 97% of the time. However, in human-annotated data(SQA3D), the distribution of *''left''* and *''right''* should be balanced.
> In contrast, our model produces a distribution that closely matches the ground truth, demonstrating the improved situated understanding ability and our generated automatic data is close to human data.
> Additionally, Table 8 in the Appendix presents a detailed breakdown of fine-tuned performance across various question types. The final EM score is the average performance across these question categories. Notably, Spartun3D demonstrates particular effectiveness in addressing *''what,"* *''is,"* and *''can"* questions.
>
>
> 2) **Performance Drops between Zero-shot and Fine-tuning.** Thank you for the question. In the zero-shot setting, Spartun3D-LLM is trained using Spartun3D, which is constructed from 3RScan, whereas SQA3D is sourced from ScanNet. The performance drop is expected due to differences in the 3D scenes between the datasets. However, compared to LEO, Spartun3D-LLM achieves an approximate 20\% improvement in the zero-shot setting, demonstrating enhanced situated understanding.
>
> 3) **Scaling Effect.** We sincerely appreciate the reviewer’s providing this constructive suggestion. In response, we conducted scaling experiments to demonstrate how model performance improves with the addition of Spartun3D datasets. Our evaluation of SQA3D shows consistent improvement as the dataset scales, underscoring the potential for further dataset expansion using our proposed method. The results are as follows, and we have included the corresponding effect curve in the new version of the paper.
> | % of Dataset | EM   |
> |--------------|------|
> | 20           | 52.7 |
> | 40           | 53.3 |
> | 60           | 53.9 |
> | 80           | 54.7 |
> | 100          | 55.0 |
>
> 4) **Clarification of L460-465.** We apologize for the confusion caused by our writing. To clarify, our intended statement is: *``We re-trained LEO exclusively on their constructed dataset from 3RScan.''*.
> LEO constructed its dataset using scenes sourced from 3RScan. However, its publicly available checkpoint was already fine-tuned on 3D tasks from ScanNet, including ScanQA, Scan2Cap, and SQA3D. To ensure a fair comparison and accurately evaluate zero-shot performance on SQA3D, we re-trained LEO exclusively on its constructed dataset from 3RScan.
> We have revised the corresponding part in our new version.
>
>
> 5) **Navigation Performance.** In Table 5, the differences between LEO and Spartun3D-LLM lie in both the training dataset and the alignment module. Specifically, The baseline here is LEO trained on its own constructed dataset and fine-tuned on other 3D tasks, excluding ObjNav. While Spartun3D-LLM is trained using the Spartun3D dataset and incorporates our specially designed spatial alignment module.
> The primary reason LEO performs poorly on navigation tasks in a zero-shot setting lies in the limited spatial information in its dataset. Unlike Spartun3D, their dataset lacks specially designed spatial information. As illustrated in the teaser (Fig.~1), when asked, ``What should you do to wash hands?``, LEO generates the answer ``sink``. While correct at an object level, this response lacks spatial context, highlighting that their model mainly emphasizes understanding objects and attributes.
> In contrast, every example in Spartun3D explicitly incorporates questions and answers involving spatial information. This design enhances the model's spatial reasoning abilities, which further helps improve the performance on downstream navigation tasks.

---

> > ### Comment · Reviewer_jrTE · 2024-11-22
> >
> > I want to thank the authors for their response. Below are my comments:
> >
> > - **How effective of automatically generated data compared to Human Data?** My question is more on the impact of data rather than the impact of spatial-alignment loss / model details. It is unclear from the table,  if the gains are coming from the data or other changes in the model design. Besides, some details are unclear -- are models in Table-8 trained on SQA3D? If so, any reasonable model should be able to learn the correct distribution -- is there a reason why it doesn't. If it is not trained on SQA3D, then maybe training on SQA3D fixes these issues as well? Additionally, while it is clear that LEO has a strong bias towards "left", Spartan3D is somewhat marginally better than LEO (1.2%), despite the ground-truth distribution being balanced. Thus, I am not sure I am seeing enough evidence yet of generated data helping significantly.
> >
> > - **Performance Drops between Zero-shot and Fine-tuning.** Sorry, which comment of mine does this answer correspond to?
> >
> > - **Navigation Performance**: Thank you for this clarification.
> >
> > - **Scaling experiment**: Thank you for this experiment.
> >
> > - **Clarification of L460-465.**: Thank you for this clarification. I don't see a revised PDF though. Also I am still confused about "I did not follow how obtaining good performance from LEO trained on Spartun3D leads to the "generalization ability of our model" conclusion? Table-3 does not show how the proposed model works with the "newly constructed 3RScan data" instead of the Spartan3D dataset."

---

> > > ### Author Response · Authors · 2024-11-24
> > >
> > > We sincerely appreciate the reviewer's prompt feedback and the opportunity to clarify our work further.
> > >
> > > **1. Questions related to Spartun3D**
> > >
> > > 1.1) **Potential Reason of Marginal Improvement for Fine-tuning Results.** Based on our analysis, the marginal improvement observed in the fine-tuning of SQA3D can be attributed to two key factors: (1) the domain gap and (2) the number of scenes.
> > > Spartun3D is constructed using 3RScan, and we mainly use approximately 300 scenes, whereas SQA3D is derived from ScanNet, which contains 650 scenes. LEO provides an experiment to show that their model trained on ScanNet struggles with generalization across 3RScan tasks, which indicates the domain gap indeed exists. Besides, the different number of scene diversity could also pose a challenge for fine-tuning, especially for tasks that rely on extensive scene variety. However, our analysis of the scaling effect suggests that expanding the dataset to include more diverse scenes could further improve performance. This finding highlights the potential of our approach to constructing larger and more diverse datasets to enhance fine-tuning results.
> > >
> > > 1.2) **Effectiveness of Spartun3D.** Although a marginal improvement in fine-tuning performance, we want to emphasize the following evidence demonstrating the effectiveness of our dataset:
> > >
> > > a) The substantial improvement (~20\%) in the zero-shot setting (second row in Table 3) compared to LEO highlights the effectiveness of our dataset. This result strongly demonstrates that our dataset enhances LEO's ability to capture situated understanding.
> > >
> > > b) In our spatial alignment module, we mainly use MSE to reduce the gap between situated spatial textual descriptions and their corresponding 3D spaces. These spatial textual descriptions are also part of our dataset. The observed improvement in the alignment module can be largely attributed to the quality and design of our dataset.
> > >
> > > c) It is evident that LEO exhibits a bias toward specific answers~(such as ``left``) based on our previous response. After training with our dataset, the model achieves a more balanced distribution of answers, especially for questions that require strong spatial reasoning and situated understanding (e.g. ``which direction`` questions discussed in Section 5.3). This balancing effect directly addresses the limitations of LEO, showcasing how our dataset encourages models to learn a more nuanced and spatially aware representation of 3D scenes.
> > >
> > > 1.3) **Clarification of Table-8.** Table-8 is trained on SQA3D, and we observe biases while we conduct experiments on the fine-tuning results.
> > >
> > >
> > > **2. Performance drops between zero-shot and fine-tuning.** This is to address the reviewer's comments that ``Additionally, just training on Spartan3D dataset results in significantly worse performance on SQA3D (by about 20\%)``  Since we train on spartan3D and test on SQA3D, we refer to this as zero-shot.
> > >
> > >
> > > **3. Clarification of L460-465.** LEO is originally trained on a mixture dataset that includes data sourced from 3RScan (LEO's constructed dataset) and ScanNet (e.g., Scannet, Scan2Cap, **SQA3D**). To ensure a fair zero-shot comparison, we train LEO on the original dataset, excluding all Scannet data~(shown in the first row of Table-3). In this whole process,
> > > **There is NO newly constructed 3RScan data** but a new version of pre-training data based solely on 3RScan for LEO.
> > >
> > > In the second row of Table 3, we fine-tune LEO with Spartun3D where Spartun3D is also constructed from the 3D scenes of the 3RScan dataset. By fine-tuning with Spartun3D, LEO achieves an approximately 20% performance improvement compared to the zero-shot LEO (i.e., the first row). This demonstrates that although LEO was only pre-trained and fine-tuned on datasets that are sourced from 3RScan, after being fine-tuned on Spartun3D with rich situated understanding tasks, the generalization ability is improved.

---

> ### Comment · Reviewer_jrTE · 2024-11-27
>
> I want to thank the authors for their response. I think this clarified a lot of mis-understandings I had. I am inclined to raised my score to 6, I still have a few gaps in my understanding.
>
> - I understand the experiment in Table-3 better now, thanks for adding a clear text on it in the paper. I think the main claim there is: LEO trained on 3RScan data they were already using does not generalize well to ScanNet based SQA3D dataset. When it is trained on the proposed Spartun3D dataset, it starts to generalize -- hence, this data is useful for training future models and would likely aid generalization. I agree with this. I do not agree, however, with the claim on Line 463-464 "generalization ability of
> our model". I do not see an evidence for Spartan3D-LLM generalizing better than LEO based on Table-3. Spartan3D-LLM is indeed better by a few percent than LEO in zero-shot setting, but it is also better than LEO on fine-tuned setting (in-domain). I think it is inconclusive whether the proposed model generalizes better or it is just stronger in general than LEO.  (Maybe by "model" the intention was to refer to the methodology of generating the dataset?)
>
> - I think the experiments with zero-shot generalization in Table-3 does not directly answer "whether the automated way of generating large amounts of data is better than human generated small-scale SQA3D data". 3RScan data is also generated automatically, and the experiment in Tabe-3 shows that their method of generating that data is worse than the proposed Spartan3D data generation. I do not expect the authors to do this experiment -- but perhaps one way to answer it is: automatically generate data on ScanNet, and compare performance on val set of SQA3D with varying amounts of generated data and human-collected data. Fine-tuning on automatic+real data and testing on val set of SQA3D can also answer this question. As the authors mention it: current fine-tuning performance does not tell us much since Spartan3D (based on 3RScan) has large domain gaps to ScanNet.
>
>
> > However, in human-annotated data(SQA3D), the distribution of ''left'' and ''right'' should be balanced. In contrast, our model produces a distribution that closely matches the ground truth, demonstrating the improved situated understanding ability and our generated automatic data is close to human data.
>
> I still do not understand this and implications drawn in 5.3. Fact 1 is: SQA3D dataset is already balanced, yet LEO shows imbalance in its output space (I am assuming that LEO is not trained on Spartan3D data). Fact 2: Spartan3D-LLM which is LEO + spatial alignment loss + trained on Spartan3D dataset does not show the bias. It is unclear if this is because of Spartan3D dataset or spatial alignment loss. I think the authors are trying to say that spatial alignment loss is not disjoint from Spartan3D dataset, as the proposed dataset helps in using the loss. But couldn't this loss also be used with original training data of LEO? if that works, then the fix to bias is the loss and not necessarily the dataset.
>
> I also don't understand the conclusion drawn by "generated automatic data is close to human data" -- SQA3D is human data and is already balanced and yet LEO has this bias. I am not sure how we are reaching at this conclusion based on the facts.

---

> ### Author Response · Authors · 2024-11-28
>
> We sincerely thank the reviewer for their prompt feedback, which has been invaluable in improving the quality of our work.
>
>
> **1) Modification of Line 463-464.** Thank you for your helpful feedback in clarifying our statement. Based on the reviewer's suggestion, we have updated the relevant sentence as follows and made the corresponding modifications in the paper.
>
> ``LEO trained on Spartun3D (LEO+Spartun3D) shows significant improvement, demonstrating the effectiveness of our dataset. Further comparisons of Spartun3D-LLM with LEO+Spartun3D demonstrate a better zero-shot learning (i.e., generalization) capability of our model.``
>
>
> **2) Generating dataset on ScanNet.**
> We sincerely appreciate the reviewer’s suggestion regarding generating data from ScanNet to address concerns about SQA3D. We agree that leveraging ScanNet could provide additional insights to address the reviewer's concern. However, a key difference between ScanNet and 3RScan is that ScanNet does not include fine-grained annotations for the objects in each scene, e.g., size, shape, state, and affordance, while such meta information about objects is crucial for applying our automatic data construction pipeline to generate high-quality and accurate situated scene graphs or situated captioning and question answering data instances. It might be achievable to obtain such information by applying various external tools but given the limited time window, we decided to leave it for future work.
>
> However, we would like to emphasize the value and broader applicability of our dataset beyond the specific scope of SQA3D. Our goal is to pre-train on a large-scale dataset, which is applicable to arbitrary tasks that need situated understanding.
> While SQA3D is a valuable resource, models trained on such human-annotated data are unable to generalize to other embodied tasks such as navigation and manipulation. In contrast, our Spartun3D dataset, including various situated tasks, enables models to excel not only in answering situated questions but also in tasks like robotic navigation in real-world environments.
>
>
> **3) Question related to Section5.3.** Thank you for pointing this out. We conducted the following Table to help the reviewer understand where the fixed bias comes from. The Table shows the number of labels for the question starting with ``which direction `` in SQA3D test dataset. LEO exhibits a noticeable bias towards ``left``, but when trained on our dataset (LEO+Spartun3D), this bias is significantly mitigated. While further adding our alignment loss~(Spartun3D-LLM) appears to further contribute to this improvement, the primary factor in addressing the bias is our dataset. We have modified Section 5.3 in the paper.
>
> | Models            | Left | Right | Forward | Backward |
> |--------------------|------|-------|---------|----------|
> | Ground-Truth | 94   | 95    | 14      | 11       |
> | LEO           | 208  | 6     | 0       | 0        |
> | LEO+Spartun3D | 78   | 130   | 3       | 3        |
> | Spartun3d-LLM | 83   | 124   | 4       | 3        |
>
>
> **4) Closer to Human Data.** We sincerely apologize for the inaccuracy in our previous rebuttal. We retract the statement and clarify that the distribution of the generated answers is more balanced and natural, better aligning with the distribution of human ground-truth labels.

---

> > ### Comment · Reviewer_jrTE · 2024-11-28
> >
> > Thank you to the authors for their response -- this clarifies my remaining doubts. I have raised my score and will be supportive of accepting this paper.

---

### Official Review · Reviewer_d7E6 · 2024-11-04

**Soundness:** 3
**Presentation:** 3
**Contribution:** 2
**Rating:** 6
**Confidence:** 4

**Summary:**

This paper introduces Spartun3D, a scalable dataset designed to enhance situated 3D understanding. The authors construct a situated scene graph to facilitate the generation of situated captions and QA pairs via LLM prompting. Additionally, they incorporate a situated spatial alignment module into a baseline 3D LLM to improve scene understanding. Experimental results demonstrate the effectiveness of their approach.

**Strengths:**

1. The situated scene graph is well-crafted and contributes to a higher-quality dataset that supports future research in situated 3D scene understanding.
2. The paper is well-written and easy to follow.

**Weaknesses:**

1. The paper could benefit from including more examples of generated data, particularly those with errors, to provide better insight into the dataset's quality. It would be valuable to discuss potential methods for correcting inaccurate captions or QA pairs generated by the LLM, as well as the impact of these errors on model performance.
2. There is a lack of a detailed ablation study on the proposed modules, specifically the Situated Textual Description and 3D Object-Text Alignment, which would help clarify their individual contributions.
3. The evaluation of object navigation relies on only four simple actions, which may weaken the findings, although it does showcase some zero-shot capabilities of the model.

**Questions:**

Please refer to the weakness section.

---

> ### Author Response · Authors · 2024-11-21
> **Response to Reviewer d7E6**
>
> We appreciate the reviewer's recognition of our contributions and the potential for future research in situated 3D scene understanding.
>
> **1) More error examples.** Thank you for highlighting this important issue. We totally agree with the reviewer that detailed error analysis is necessary. We randomly sampled 50 examples from each task (200 in total) to validate the quality of our automatically generated data and manually assess the quality from the aspects of language naturalness and spatial fidelity. Language naturalness evaluates whether the generated texts are natural or written by a human, while spatial fidelity ensures that data accurately reflects the 3D scene. We observe 26 errors in total and summarize them into the following categories.
>
> - **Semantic Errors**: The generated sentences may contain semantic mistakes. For instance, the answer ``You should go in front of you to water the potted plant.``
> - **Common Sense Violations**: The generated content may occasionally conflict with basic common sense knowledge, such as producing unusual questions and answers. For example, it might generate a question like ``If I want to store items, which object is most accessible?`` with the answer being ``trash bin.`` This issue arises because the human-annotated data includes an affordance for trash bins as objects for storing items. Such annotations inadvertently influence GPT-4o to generate QA pairs that conflict with common sense knowledge.
> - **Spatial Inconsistencies**: Errors in capturing or reasoning about spatial relationships in the 3D environment primarily occur in Situated Planning tasks, which demand complex two-step spatial reasoning. These errors often arise because the second action depends on the outcome of the first action, and inaccuracies sometimes occur during the generation of the second action.
> - **Misalignment between visual context and textual descriptions**: In some cases, the agent's view is obstructed due to the room layout or object size. For example, consider the situation: ``Standing beside the sofa, there is a closet on your left.`` However, the closet is actually located in another room and cannot be seen from the agent's current standpoint. To address this issue, we designed a scenario where the agent stands beside a pivot object and consistently faces the center of the pivot object rather than facing a random object that could potentially be obstructed. Additionally, we incorporated pass-by spatial information to enhance the agent's awareness of surrounding objects, providing a more comprehensive sense of the environment.
>
> The following table presents detailed statistics of the errors for each category of generated text. Certain errors are task-specific; for instance, spatial inconsistencies predominantly occur in situated planning. Similarly, misalignment issues are more common in situated captioning, as captions often include descriptions of surrounding objects, which increases the likelihood of mentioning obstructed objects.
> | Error Type              | Captioning | Attr. and Rel. | Affordance | Planning |
> |-------------------------|------------|----------------|------------|----------|
> | Semantic Errors         | 0          | 2              | 2          | 0        |
> | Common Sense Violations | 0          | 3              | 5          | 2        |
> | Spatial Inconsistencies | 0          | 0              | 0          | 4        |
> | Misalignment            | 7          | 0              | 0          | 1        |
>
> Finally, we also sample 80 answers~(20 for each type) predicted by our model and confirm that the model is not really affected by the little noise contained in the data. As shown in the following second table, Errors related to semantics, common sense, and misalignment errors are much less than the noise in training. However, spatial inconsistencies were observed. We attribute these spatial inconsistency errors to the inherent difficulty of spatial reasoning rather than to noise in the training data. This conclusion is supported by the observation in the two tables: during training, spatial errors occur primarily in planning. However, during prediction, spatial errors occur uniformly across all types of tasks.
> | Error Type               | Captioning | Attr. and Rel. | Affordance | Planning |
> |--------------------------|------------|----------------|------------|----------|
> | Semantic Errors     | 0          | 0              | 0          | 0        |
> | Common Sense Violations | 0       | 0              | 1          | 0        |
> | Spatial Inconsistencies | 3       | 2              | 4          | 2        |
> | Misalignment         | 2          | 0              | 0          | 0        |

---

> > ### Author Response · Authors · 2024-11-21
> >
> > **2) Ablation Study on Proposed Module.** Thank you for your question. In fact, situated textual descriptions and 3D object-text alignment are not separate modules; rather, the situated textual descriptions serve as supervision to train the 3D object-text alignment. Therefore, these two components coexist and function as an integrated module~(Situated Spatial Alignment Module), ablation and separating these components is not relevant in this case. The ablation study for the Spartun3D dataset and Situated Spatial Alignment Module has also been shown and discussed in Tables 2 and 3: LEO*+Spartun3D v.s. Spartun3D-LLM*.
> >
> > **3) Simple Action in Object Navigation.** We evaluated navigation performance on the standard ObjNav dataset using its predefined action set of four actions. These four actions are commonly employed in navigation tasks, such as Vision and Language Navigation[1][2]. While the action space is simple, the intermediate reasoning required remains complex. The agent must understand instructions, perceive the 3D visual environment, history actions and reason effectively to generate action, making this task challenging.
> >
> > [1]Anderson, Peter, et al. "Vision-and-language navigation: Interpreting visually-grounded navigation instructions in real environments."
> >
> > [2]Krantz, Jacob, et al. "Beyond the nav-graph: Vision-and-language navigation in continuous environments."

---

> > > ### Comment · Reviewer_d7E6 · 2024-11-26
> > >
> > > Thank you for your response, which has addressed most of my concerns. Based on this, I will raise my score from 5 to 6.

---

### Author Response · Authors · 2024-11-24
**General Response**

We sincerely appreciate the reviewers' constructive comments and feedback on our work. We are grateful that all reviewers acknowledged the motivation behind our research and its contributions to the community, as well as the well-organized presentation of our work.

At the same time, we have carefully addressed the valuable suggestions provided to improve our paper. Below, we summarize the key revisions made in our paper (labeled as blue front):

- **Human Evaluation (Reviewers d7E6, D9vh, and kEmt)**: We have included human evaluation results in Section 3.4, assessing the quality of Spartun3D in terms of language naturalness and spatial fidelity.
- **Scaling Effect (Reviewers jrTE and kEmt)**: We have added an analysis of the scaling effect in Section 5.3 to highlight the impact of dataset expansion on performance.
- **Clarification of SQA3D Performance (Reviewer jrTE)**: We have clarified the explanation of SQA3D performance in Section 5.2 to address the reviewer's concerns.
- **Error Analysis (Reviewer d7E6)**: Updates related to error analysis can be found in Appendix A.1, providing further insights into the error categories.

We hope that these revisions address the reviewers’ concerns and provide a stronger foundation for re-evaluating our submission. If there are any additional questions or comments, we are happy to address them.

---

### Author Response · Authors · 2024-11-25
**General Response**

Dear Reviewers,

We sincerely appreciate the time and effort you’ve devoted to reviewing our work. We understand that your schedule may be quite busy, and we are truly grateful for your valuable feedback. As the Author-Reviewer discussion phase is extended, we would greatly value the opportunity to engage in further discussion with you during this period of time. Our aim is to gain insights into whether our responses effectively address your concerns and to ascertain if there are any additional questions or points you would like to discuss.

We look forward to the opportunity to discuss this further with you. Thank you for your thoughtful consideration.

Best regards,

Authors

---

### Author Response · Authors · 2024-12-04
**General Response**

Dear Area Chair and Reviewers,

We would like to express our sincere gratitude for your efforts in facilitating the discussion regarding our paper.
We sincerely thank all reviewers (d7E6, jrTE, D9vh, kEmt) for recognizing the **innovative contribution of our dataset** in addressing current limitations in 3D-based LLMs for **situated understanding**. We are grateful to reviewers jrTE and D9vh for acknowledging **the effectiveness of our alignment strategy** and to reviewers D9vh and kEmt for highlighting **the strengths of our comprehensive experiments and practical applications.** Additionally, we appreciate all reviewers' feedback affirming that our paper is **well-written, clear, and sound**.


We also appreciate the reviewers' constructive suggestions to improve the quality of our work. Below, we summarize the key points addressed in our discussion:

**1) Quality Control and Error Analysis of Spartun3D:**
We appreciate the reviewers highlighting this common concern (d7E6, D9vh, and kEmt) to improve the quality of our paper.  Although we included some human evaluation in the original submission, we agree with the reviewer on the need for deeper evaluation and error analysis.
To address this, we revised Section 3.4 to include comprehensive human evaluations focusing on both language naturalness and spatial fidelity.  Additionally, we analyzed how different prompting strategies influence the dataset quality. We also provided the error analysis in Appendix A.1.
Following these updates, reviewers d7E6 and kEmt confirmed that their concerns had been addressed.
Although reviewer D9vh did not participate in the discussion period, we believe our new experiments have addressed the reviewer's main concern as well.

**2) Scaling Effects:** We sincerely appreciate the reviewers (jrTE and kEmt) for highlighting the need for scaling effect experiments, which we agree are essential to include. To address this concern, we have added a scaling effect experiment in Section 5.3. Both reviewers, jrTE and kEmt, have confirmed that we have addressed their concerns.


**3) Clarification of SQA3D Performance:** We sincerely thank Reviewer jrTE for this valuable feedback, which helped us identify and address the writing issues that caused confusion regarding our experimental results. We have revised the relevant part in Section 5.2 to clarify the effectiveness of the dataset and our designed alignment module. Reviewer jrTE has confirmed that these doubts have been resolved and expressed support for accepting our paper.

We have carefully refined our work and incorporated all suggested improvements in the revised submission, with updates clearly marked in blue. These changes significantly enhance the clarity and completeness of our paper. We thank the reviewers and area chair for their time and valuable suggestions in helping us improve this work.

Best,

Authors

---

### Meta-Review · Area_Chair_2Wgk · 2024-12-20

**Metareview:**

**Summary**

The paper aims to study the ability of 3D-LLMs to perform situated spatial reasoning, where given 3D scene and an agent's location and pose, the agent needs to either provide descriptions of surrounding objects (situated captioning) or answer questions (situated question-answering).  To do so, the paper introduces Spartun3D, a situated 3D dataset consisting of 133K examples generated using LLMs for evaluating captioning and QA.  The paper also proposes Spartun3D-LLM, which adds a situated spatial alignment module to a recent 3D-based LLM (LEO). Experiments show that the proposed module improves performance on the proposed Spartun3D benchmark as well as other 3D language tasks.

The main contributions is the Spartun3D dataset, the proposed situated spatial alignment module, and the experiments demonstrating the effectives of the proposed approach.

**Strengths**

Reviewers noted the following strengths of the work:
- The dataset is useful for the community [d7E6,jrTE,D9vh]
- Reviewers found the paper to be well-written and easy to follow [d7E6,jrTE,kEmt].  The AC also find the problem formulation, dataset, and proposed module to be clearly described.
- Experiments show proposed module is effective [jrTE,D9vh]
- Tasks investigated by this work is underexplored [kEmt]

**Weaknesses**

Reviewers noted the following weaknesses:
- Concerns about the use of LLM to generate the data [jrTE,D9vh,kEmt]  including
  - Quality of the data compared to human generated data
  - Whether scaling the data up actually can help train better models
- Some aspects where not initially clear [jrTE,kEmt]
- The evaluation can include more tasks and recent methods [D9vh]
- Missing discussion of computational costs [D9vh]
- Reviewer also requested additional details such as:
  - More examples of generated data [d7E6]
  - Detailed ablation study [d7E6]

The main common concern across reviewers was the quality of the generated data and request for additional information on how scaling the data affects the model performance.  This concern (as well as other concerns) where addressed by the authors during the author response period.

**Recommendation**

Overall reviewers are slightly positive on this work.  The AC believes the dataset can be useful, and the paper is clear and so recommends acceptance.

**Additional Comments On Reviewer Discussion:**

Initially all reviewers were slightly negative on the work (with a score of 5).  After the author response period, three of the reviewers increased their scores to 6 (marginally positive) as the authors did a good job of responding to reviewer concerns and updating the manuscript.  The authors provided human evaluation of the data, additional experiments about scaling, provided error examples and improved writing for sections that were confusing.

The last reviewer (D9vh) did not engage in discussion and did not update their score.  The AC feels most of this reviewer has been answered by the author response.  One weakness noted by the reviewer that was not fully addressed was evaluation on more embodied tasks (the authors included one manipulation task, maybe there could be more) and more baselines (the authors provided an explanation of why LEO was selected).  Nevertheless, the AC feels the evaluation was sufficient (there can always be more tasks and baselines) and thus recommend acceptance.

---

### Decision · Program_Chairs · 2025-01-22

Accept (Poster)